# Protein:Protein interactions in the cytoplasmic membrane apparently influencing sugar transport and phosphorylation activities of the *e. coli* phosphotransferase system

**Mohammad Aboulwafa** [1,2], **Zhongge Zhang**[1], **Milton H. Saier, Jr.** [1] *

**1** Department of Molecular Biology, Division of Biological Sciences, University of California at San Diego, La Jolla, CA, United States of America, **2** Department of Microbiology and Immunology, Faculty of Pharmacy, Ain Shams University, Abbassia, Cairo, Egypt

* msaier@ucsd.edu

**Data Availability Statement:** All relevant data are within the manuscript and its Supporting Information files.

## Abstract

The multicomponent phosphoenolpyruvate (PEP)-dependent sugar-transporting phospho-transferase system (PTS) in *Escherichia coli* takes up sugar substrates from the medium and concomitantly phosphorylates them, releasing sugar phosphates into the cytoplasm. We have recently provided evidence that many of the integral membrane PTS permeases interact with the fructose PTS (FruA/FruB) [1]. However, the biochemical and physiological significance of this finding was not known. We have carried out molecular genetic/biochemical/physiological studies that show that interactions of the fructose PTS often enhance, but sometimes inhibit the activities of other PTS transporters many fold, depending on the target PTS system under study. Thus, the glucose (Glc), mannose (Man), mannitol (Mtl) and N-acetylglucosamine (NAG) permeases exhibit enhanced *in vivo* sugar transport and sometimes *in vitro* PEP-dependent sugar phosphorylation activities while the galactitol (Gat) and trehalose (Tre) systems show inhibited activities. This is observed when the fructose system is induced to high levels and prevented when the *fruA/fruB* genes are deleted. Overexpression of the *fruA* and/or *fruB* genes in the absence of fructose induction during growth also enhances the rates of uptake of other hexoses. The β-galactosidase activities of *man*, *mtl*, and *gat-lacZ* transcriptional fusions and the sugar-specific transphosphorylation activities of these enzyme transporters were not affected either by frustose induction or by *fruAB* over-expression, showing that the rates of synthesis of the target PTS permeases were not altered. We thus suggest that specific protein-protein interactions within the cytoplasmic membrane regulate transport *in vivo* (and sometimes the PEP-dependent phosphorylation activities *in vitro*) of PTS permeases in a physiologically meaningful way that may help to provide a hierarchy of preferred PTS sugars. These observations appear to be applicable in principle to other types of transport systems as well.

**Funding:** This work was supported by U.S. National Institutes of Health, MS - GM077402, https://www.nih.gov/. The funders had no role in study design, data collection and analysis, decision to publish, or preparation of the manuscript.

**Competing interests:** The authors have declared that no competing interests exist.

## Introduction

The prokaryotic phosphoenolpyruvate (PEP):sugar phosphotransferase system (PTS) consists of two general energy coupling proteins, Enzyme I, (EI, PtsI; TC# 8.A.7.1.1 in the Transporter Classification Database, TCDB (www.tcdb.org)) and HPr (HPr, PtsH; TC# 8.A.8.1.1), as well as the sugar-specific Enzyme II (EII) complexes (IIABC(D); TC families 4.A.1-7) [1–5]. The EII complexes usually consist of three proteins or protein domains, IIA, IIB and IIC, although the EII systems of one family, the mannose EII family, consist of four proteins, IIA, IIB, IIC and IID, where IID is an integral membrane protein required for the function of the IIC transporter [6, 7]. Moreover, several members of the glucose family (TC# 4.A.1) use the same EIIA$^{Glc}$ protein (Crr).

The IIA and IIB proteins are cytoplasmic phosphoryl carrier proteins or protein domains while the IIC proteins/domains are integral membrane transporters that catalyze sugar phosphorylation concomitantly with sugar uptake into the cell [8, 9]. In these coupled sugar transport/phosphorylation reactions, called "group translocation," the phosphoryl moiety of IIB-P is transferred to the incoming sugar to yield a cytoplasmic sugar phosphate (Sugar-P) [10]. The overall phosphoryl transfer reactions are shown in Fig 1.

In this Figure, arrows indicate the pathways of phosphoryl transfer between the protein constituents of the PTS as defined in the introductory paragraph. Essentially, the same general phosphoryl transfer pathway is used for the phosphorylation of all PTS sugars such as glucose (Glc) and its non-metabolizable analogue, methyl α-glucoside (αMG), transported by the glucose PTS (IIA$^{Glc}$/IIB$^{Glc}$/IIC$^{Glc}$, Crr/PtsG; TC# 4.A.1.1.1), mannose (Man) and its non-metabolizable analogue, 2-deoxyglucose (2DG), transported by the mannose PTS (ManXYZ; IIA$^{Man}$-IIB$^{Man}$/IIC$^{Man}$/IID$^{Man}$; TC# 4.A.6.1.1), trehalose (Tre; an α, α-disaccharide of glucose), transported by the trehalose PTS (Crr/TreB; IIA$^{Glc}$/IIBC$^{Tre}$; TC# 4.A.1.2.4), and galactitol, transported by the galactitol PTS (IIA$^{Gat}$/IIB$^{Gat}$/IIC$^{Gat}$ = GatA/GatB/GatC). Many other PTS sugars

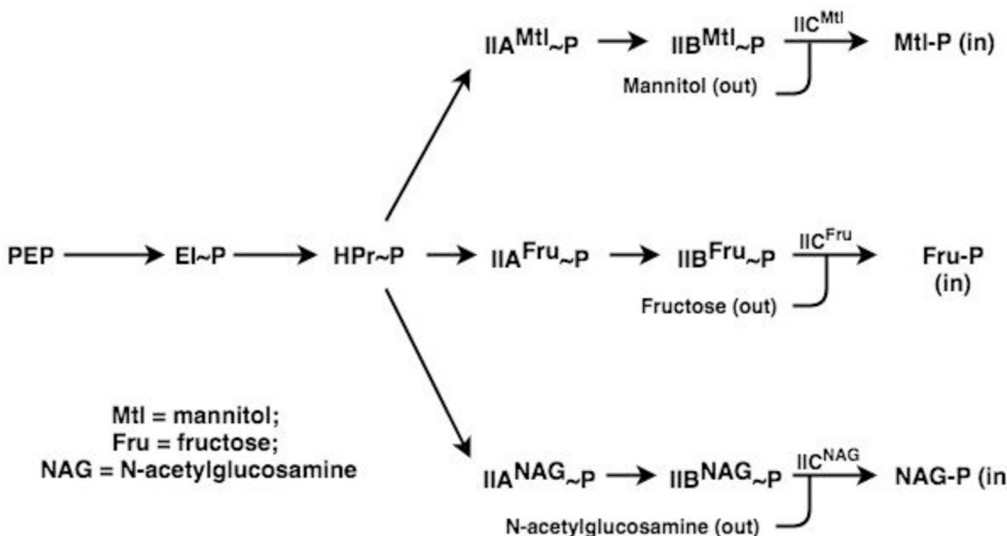

**Fig 1. Phosphoryl transfer pathway for the bacterial phosphotransferase system (PTS).** The figure shows the transfer of a phosphoryl group from phosphoenolpyruvate (PEP) to enzyme I (EI), then to HPr, then to the sugar-specific enzyme IIAs and then to the enzyme IIBs before transfer to the incoming sugar via the enzyme IIC, which catalyzes both transport and phosphoryl transfer in a coupled process. The PTS also catalyzes the group translocation of many other sugars. As indicated, the three extracellular sugars represented are mannitol (Mtl), fructose (Fru), and N-acetylglucosamine (NAG), and their phosphorylated derivatives are released into the cytoplasm. Arrows indicate the pathways of phosphoryl transfer.

are transported and phosphorylated in a coupled transport/phosphorylation process known as group translocation (see Class 4 in TCDB).

In addition to these three D-sugars, mannitol (Mtl), fructose (Fru) and N-acetylglucosamine (NAG), the PTS transports and phosphorylates many other sugars, and each bacterial or archaeal species possessing the PTS has a different complement of PTS Enzyme II complexes. Sugars transported via the PTS in various organisms include aldo- and keto-hexoses, amino sugars and their N-acetylated derivatives, hexitols, pentoses, pentitols and a variety of disaccharides, oligosaccharides and glycosides [8]. In *Escherichia coli*, there are many Enzyme II (EII) complexes, some of which are still not functionally characterized [11]. The EIIs we will be concerned with in this report, in addition to the three illustrated in the scheme shown in Fig 1, are specific for sugars such as galactitol (Gat), glucose (Glc) and the non-metabolizable glucose analogue, methyl α-glucoside (αMG), trehalose (Tre), and mannose (Man) [this system also transports Glc, 2-deoxyglucose (2DG), glucosamine (Glm) and fructose]. These systems are tabulated in Table 1 with their protein abbreviations, protein domain orders, TC numbers (TC #s) in the Transporter Classification Database (TCDB; www.tcdb.org), and primary sugar substrates [12–15].

In addition to the transport reaction that can be studied *in vivo*, there are two *in vitro* sugar phosphorylation reactions, the PEP-dependent sugar phosphorylation reaction that depends on EI, HPr and the complete EII complex, and a sugar-P:sugar transphosphorylation (TP) reaction that depends only on IIB and IIC [16–19]. All of these reactions have been used to investigate the consequences of integral membrane protein interactions in the current study.

Recently, Babu et al published global interactome data for *E. coli*, including cytoplasmic proteins, integral inner and outer membrane proteins and periplasmic proteins [1]. These studies revealed that many Enzyme II components interact with other soluble and integral membrane proteins of the PTS as well as with other non-PTS proteins. A representative selection of these PTS protein interactions is reproduced in Table 2. Of all the Enzyme II constituents, the fructose PTS proteins, FruA and FruB, appeared to form the most extensive PTS interactome network.

The observation that the Fru PTS might serve to coordinate the activities of other PTS EII complexes correlated with the previously published suggestion that the fructose PTS was the primordial PEP-dependent PTS Enzyme II complex [20]. This suggestion resulted from several observations. (1) The fructose PTS, of all the sugar-specific Enzyme II complexes, is most widely distributed in the prokaryotic world. (2) The fructose system in *E. coli* has more fructose-like Enzyme II complex homologs than any other. (3) Fructose is the only sugar that feeds directly into glycolysis without interconversion to another sugar derivative. (4) Only fructose can feed into glycolysis via two distinct PTS-mediated pathways: Fru → Fru-1-P → Fru-1,6-bis-P ($II^{Fru}$), and Fru → Fru 6-P → Fru-1,6-bis P ($II^{Man}$). (5) Only the fructose PTS of

**Table 1. Proteins of the PTS in *E. coli*, relevant to the study reported here.** References for these systems can be found in TCDB.

| PTS Complex | Protein Constituents (Domains) | TC # | Substrate(s) |
|---|---|---|---|
| Fructose (Fru) | FruA: $IIC^{Fru}$-$IIB^{Fru}$-$IIB^{,Fru}$<br>FruB: IIA-FPr | 4.A.2.11 | Fru |
| Mannitol (Mtl) | MtlA: $IIC^{Mtl}$-$IIB^{Mtl}$-$IIA^{Mtl}$ | 4.A.2.1.2 | Mtl |
| Galactitol (Gat) | GatA: $IIA^{Gat}$; GatB: $IIB^{Gat}$; GatC: $IIC^{Gat}$ | 4.A.5.1.1 | Gat |
| N-Acetylglucosamine (NAG) | NagE: $IIC^{NAG}$-$IIB^{NAG}$-$IIA^{NAG}$ | 4.A.1.1.2 | NAG |
| Glucose (Glc) | PtsG: $IIC^{Glc}$-$IIB^{Glc}$;<br>Crr: $IIA^{Glc}$ | 4.A.1.1.1 | Glc, methyl α-glucoside (αMG) |
| Mannose (Man) | ManX: $IIAB^{Man}$; ManY: $IIC^{Man}$; ManZ: $IID^{Man}$ | 4.A.6.1.1 | Man, 2-deoxyglucose (2DG), Glc, Fru, glucosamine (Glm) |
| Trehalose (Tre) | TreB: $IIB^{Tre}$-$IIC^{Tre}$<br>(Uses $IIA^{Glc}$; there is no $IIA^{Tre}$) | 4.A.1.2.4 | Tre |

**Table 2. Selected high scoring PTS transporter (Enzyme II complex) interactions.** The data were derived from Babu et al., 2018 [1]. The higher the score, the stronger the interaction should be, although these scores are also dependent on the protein concentrations.

| Protein 1 | Protein 2 | Score |
|-----------|-----------|-------|
| FruA | FruB | 12.9 |
| FruA | MtlA | 11.4 |
| FruA | GatC | 10.3 |
| FruA | NagE | 9.4 |
| FruA | TreB | 7.4 |
| FruA | MngA | 7.1 |
| FruA | DhaL | 6.9 |
| MtlA | GatC | 9.7 |
| MtlA | FruB | 8.4 |
| ManY | GatA | 7.5 |
| ManY | GatB | 6.9 |
| ManY | GatC | 7.4 |
| ManY | MngA | 7.7 |
| NagE | PtsG | 7.0 |
| NagE | FruB | 5.4 |

functionally characterized systems in *E. coli* has its own HPr-like protein, FPr. (6) Only the fructose Enzyme II complex of *E. coli* and many other bacteria have extra IIB domains that function in protein-protein interactions rather than phosphoryl transfer [21].

More recent studies revealed that there are at least three evolutionarily distinct superfamilies of PTS Enzyme IIC components [22]. First, the PTS Glucose-Glucoside (Glc) family, the Fructose-Mannitol (Fru) family, the Lactose-N, N′-Diacetylchitobiose (Lac) family, and (4) probably the Glucitol (Gut) family comprise one large sequence-divergent superfamily with all IIC constituents being homologous (TC# 4.A.1-4; see TCDB, www.tcdb.org) [12, 13, 23, 24]. Second, the Galactitol (Gat) family (TC# 4.A.5) and the L-Ascorbate (L-Asc) family (TC# 4.A.7) comprise a distinct superfamily [11, 24, 25], and finally, the functionally diverse Mannose family (TC# 4.A.6) comprises the third PTS superfamily [26]. Each of these three superfamilies of IIC permeases are believed to have evolved independently of each other [22, 27]. However, it should be noted that the IIA and IIB phospho-carrier constituents, which also probably evolved at least 3 times independently of each other, did not coevolve with the IIC permease constituents, suggesting that there has been extensive shuffling of the IIA, IIB and IIC constituents/domains during the evolutionary divergence of these protein complexes [22, 28–30].

Bacterial cytoplasmic membranes are crowded with integral membrane proteins that form complexes, inhibiting their lateral mobilities, constraining their flexibilities and probably altering their activities [31–34]. Our integral membrane interactome data suggested that many membrane proteins interact with many other membrane proteins, with a wide range of affinities [1]. Some of the high scoring interactions are likely to be permanent, high affinity interactions, while lower scoring interactions are more likely to be transient in nature [35, 36]. Among the observed interactions were many involving Enzyme IIC constituents of the PTS, particularly the fructose IIC protein, FruA, which also bears two additional domains, IIB and IIB′, where IIB′ is not involved in phosphoryl transfer, but plays a role in protein-protein interactions [21] (Table 2).

As noted above, the PTS offers technical advantages over other transport systems for its characterization. It can be assayed *in vivo* by measuring its sugar transport (uptake) activities, and it can be assayed *in vitro* by measuring its sugar phosphorylation reactions. These include its PEP-dependent sugar phosphorylating activities as well as its sugar-P-dependent

transphosphorylating (TP) activities [19, 37]. The latter reactions require only the IIB and IIC domains which are usually, but not always, fused together (see Table 1). Thus, while the PEP: sugar activity requires protein:protein interactions involving EI, HPr, IIA and IIBC, the sugar-P:sugar TP reaction does not depend on these interactions (except for II$^{Man}$) since only the IIC and IIB domains are directly involved.

The results of the studies reported here suggest that in the cytoplasmic membrane of *E. coli*, the transport of various PTS sugars is influenced by the presence of other PTS permeases in general, and the degree of activation or inhibition depends on the specific systems under study as well as their concentrations in the membrane. Thus, high level expression of the fructose EII complex activates the mannitol, N-acetyl glucosamine, glucose and mannose systems, but it inhibits the galactitol and trehalose systems in wild type cells. Although these protein-protein interactions do not appreciably affect the sugar-P:sugar TP reactions or synthesis of the target EIIs, we did observe a highly specific activation of the *in vitro* PEP-dependent phosphorylation of mannitol and N-acetylglucosamine, dependent on both FruA and FruB. In this case, we propose that FruB activates MtlA, but that this activation depends on the presence of FruA which could anchor FruB to the membrane, adjacent to MtlA. The more general conclusion is that integral membrane transport proteins in the cytoplasmic membranes of *E. coli* are in close contact and influence each other's activities in the intact cell, either in one direction (activation) or the other (inhibition). In some cases, these effects appear to be highly specific, while in others they may be more general. This is, to the best of our knowledge, the first example where global proteomic analyses based on a complete interactome data set, have led to the discovery of regulating interactions influencing the enzymatic and transport activities of some of the interacting integral membrane proteins.

## Results

The *E. coli* integral membrane protein interactome, which includes thousands of putative interactions with other integral membrane proteins as well as other cellular proteins, has been published [1]. Table 2 summarizes the high scoring interactions observed between the constituents of different Enzyme II complexes of the PTS. Not surprisingly, the fructose-1-P-forming fructose-specific membrane constituent of the PTS, FruA (the enzyme IICBB'$^{Fru}$), interacts with its own FruB protein, the fructose-specific enzyme IIA-FPr protein, the immediate phosphoryl donor for fructose transport energization and phosphorylation, thus providing the energy for fructose uptake via FruA (Table 1; [21]). The interaction score is high (12.9), indicating that this interaction occurs with high affinity since the conditions used to measure these interactions involved growth in LB medium without induction of PTS gene expression [1]. In addition, FruA appeared to interact with several other Enzyme II constituents of the PTS with large scores, far more than for any other PTS Enzyme IIC (Table 2). These interactions include those with MtlA (the IICBA specific for mannitol; score of 11.4), GatC (The galactitol Enzyme IIC; score of 10.3), NagE (The N-acetylglucosamine Enzyme IICBA; score of 9.4), TreB (The trehalose Enzyme IIBC; score of 7.4), and MngA (the 2-0-α-mannosyl-D-glycerate Enzyme IIABC; score of 7.1) (Tables 1 and 2). Other Enzymes II appeared to interact with FruA with lower scores. MtlA additionally interacts with FruB (score, 8.4) and GatC (score, 9.7), while the broad specificity glucose/mannose/fructose/glucosamine Enzyme IIC, ManY, interacts with all three constituents of the galactitol Enzyme II complex, GatA (7.5), GatB (6.9) and GatC (7.4) as well as MngA (7.7) (Table 2).

Many other integral membrane proteins, several of which proved to be transporters, also appeared to interact with FruA. These include the MDR pumps, AcrB (score of 12.9) and MdtF (17.9), the dicarboxylate transporter, DcuA (17.2), the iron porters, FeoB (11.4) and Fet

(YbbM; 14.3), the FocA formate channel (9.5), the NupC nucleoside uptake porter (19.6), the MgtA Mg$^{2+}$ uptake ATPase (12.1) and others [1]. These results suggest that numerous integral membrane transport systems interact, possibly within the hydrophobic matrix of the cytoplasmic membrane, but their biochemical and physiological significance could not be determined from these data alone. Therefore, the transport studies and biochemical analyses reported here were conducted, specifically for the putative PTS protein interactions.

## Confirmation of the interactome results using a bacterial two hybrid system

As reported by Babu et al. [1], FruA interacts with other membrane proteins as well as its cognate cytoplasmic partner, FruB (Table 2). A bacterial adenyl cyclase two hybrid (BACTH) system was used to confirm several of these interactions, conducted according to the manufacturers' instructions (Euromedex bacterial two-hybrid system, Cat # EUK001). Interactions of FruA were demonstrated for the GatC, NagE, TreB and MtlA membrane proteins as well as the cytoplasmic FruB protein. Also, interactions of FruB were observed for MtlA and NagE (S1 Fig). These observations confirm some of the interactions reported by Babu et al., 2018 [1] and suggest that many integral membrane proteins interact with each other in the plane of the membrane, many of them, most likely, in a transient fashion.

## Transport studies involving PTS group translocators

Because the FruA protein seemed to interact with many other Enzyme II protein complexes with high scores, more than for any of the other PTS proteins, we decided to examine the effect of growth with and without fructose to induce synthesis of the fructose-specific Enzyme II complex, FruA/FruB, on the rates of uptake of other PTS sugars in wild type (WT) *E. coli* cells. The results are summarized in Fig 2A, which gives the relative apparent rates of sugar transport by the fructose-induced cells relative to uninduced cells. The raw data for this Figure are presented in S1, S2, S3, S3B, S4, S4B, S5 and S5B Tables. S3B, S4B and S5B Tables are controls for S3, S4 and S5 Tables, respectively. For example, in S3 and S4 Tables, fructose was present during growth, but in S3B and S4B Tables, fructose was absent. Clearly, induction by fructose was necessary for regulation; values recorded represent actual values within experimental error, as indicated by the standard deviation values. As an additional control, the same experiments were conducted with the isogenic strain deleted for the *fruBKA* operon (*ΔfruBKA*; triple mutant (TM)), Fig 2B.

The results presented in Fig 2A show that in a wild type *E. coli* strain, growth in the presence of fructose enhanced the amounts of uptake of some radioactive sugars while decreasing the amounts of uptake of others. Thus, the apparent mannitol (Mtl) transport rate increased 7-fold, the apparent N-acetylglucosamine (NAG) transport rate increased 6.4-fold, 2-deoxyglucose (2DG) uptake increased 8.7-fold, and methyl α-glucoside (αMG) uptake increased 2.8-fold, due to the presence of fructose in the growth medium. Interestingly, the apparent uptake of trehalose (Tre) and galactitol (Gat) decreased while galactose (Gal, a non-PTS sugar) uptake did not change appreciably.

The fructose operon was completely deleted, and this *ΔfruBKA* deletion strain (triple mutant; TM) was used in parallel experiments (Fig 2B). With the *fruBKA* operon absent, the apparent fructose transport rate was reduced to 5% of the wild type rate. None of the other apparent PTS sugar uptake rates, except for that of 2-deoxyglucose, taken up by the mannose system, was affected by the inclusion of fructose in the growth medium. This last fact may have resulted because fructose is a (poor) substrate of the mannose EII complex, and therefore may also be an inducer of this system.

When the TM was examined relative to the wild type (WT) strain, with only LB in the growth medium, none of the apparent sugar uptake rates was affected (Fig 2C), but when

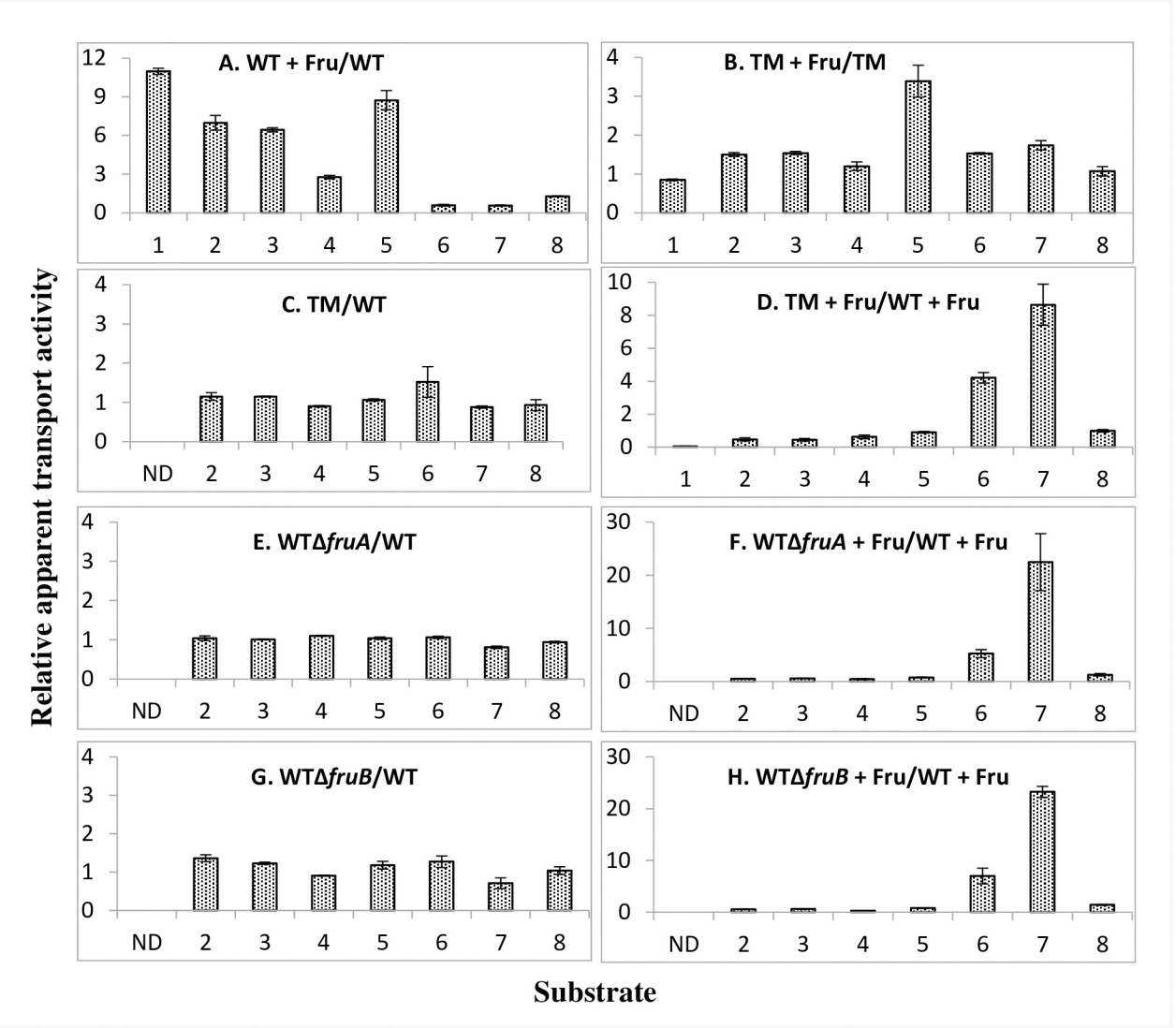

**Fig 2. Ratios of apparent transport activities for various sugars (PTS substrates and galactose) by wild type *E. coli* BW25113 (WT) and its *fruBKA* isogenic mutants (BW25113Δ*fruA*, BW25113Δ*fruB* and BW25113-*fruBKA:kn* (triple mutant, TM)) using cells grown in LB with (WT or mutant + Fru) and without 0.2% fructose.** (A) Relative apparent sugar uptake by wild type *E. coli* BW25113 (WT) grown in the presence and absence of fructose. (B) Relative apparent sugar uptake by the *fruBKA* triple *E. coli* BW25113 mutant (TM) grown in the presence and absence of fructose. (C&D) Relative apparent sugar uptake by the triple mutant (TM) relative to that of the wild type *E. coli* BW25113 strain when grown in LB or LB plus 0.2% fructose, respectively, as indicated. (E&F) Relative apparent sugar uptake by a *fruA* mutant relative to that of the wild type *E. coli* BW25113 strain when grown in LB and LB plus 0.2% fructose, respectively. (G&H) Relative apparent sugar uptake by a *fruB* mutant relative to that of the wild type *E. coli* BW25113 strain when grown in LB or LB plus 0.2% fructose, respectively. 1, Fructose; 2, Mannitol; 3, N-acetylglucosamine; 4, Methyl alpha glucoside; 5, 2-Deoxyglucose; 6, Trehalose; 7, Galactitol and 8, Galactose. The raw data for these plots are shown in supplementary materials (S1, S2, S3, S3B, S4, S4B, S5 and S5B Tables). Note: In this Figure and elsewhere, αMG and 2DG uptake values represent accumulation levels, while for other sugars, a combination of uptake + metabolic rates were estimated. Galactose is taken up via the GalP secondary carrier, not by the PTS.

fructose was present during growth, the apparent transport of those sugars that had been stimulated by fructose induction in the WT showed low ratios of uptake, while those that were inhibited in the WT showed increased ratios (Fig 2D), in agreement with those presented in Fig1A. When either *fruA* or *fruB* was deleted, all regulatory effects were abolished (Fig 2E and 2G), as was true for the Δ*fruBKA* strain (Fig 2C). However, the apparent transport ratios, Δ*fruA*/WT and Δ*fruB*/WT, showed depressed ratios for those sugars whose uptake was

increased by *fru* operon induction in the WT (Mtl, NAG, αMG and 2DG), and increased values for those sugars whose apparent transport was depressed by *fru* operon induction (Tre and Gat) (compare Fig 2A with Fig 2D, 2F and 2H).

In summary, it appears that high level expression of the *fruBKA* operon has effects on the uptake of other PTS sugars, with apparent Mtl, NAG, αMG and 2DG transport consistently showing activation while those for Gat and Tre show inhibition. In *ΔfruA* and *ΔfruB* mutants, no regulatory effects were observed. These results suggested that either FruA, FruB, or more likely, both, serve(s) regulatory roles, influencing the apparent rates of transport of other PTS sugars, positively or negatively, depending on the sugar-specific transport system assayed.

Fig 3 presents corresponding results for strains in which the *fruA*, *fruB*, or *fruA* and *fruB* genes were overexpressed. Overexpression of *fruA* in the TM genetic background increased apparent Mtl and NAG transport rates 4-5-fold, but increased apparent αMG and 2DG transport rates 7 and 11-fold, respectively, compared to the control strain bearing the empty pMAL plasmid (Fig 3A). When the same experiment was conducted in the wild type genetic background, qualitatively similar results were obtained, but enhancement of the apparent transport rates were diminished except for the mannose/2DG uptake system, which was increased (Fig 3B). Only in the WT background did overexpression of *fruA* or *fruB* increase the apparent 2DG uptake rate. Surprisingly, in both cases, apparent Tre and Gat transport rates were not appreciably depressed, possibly due to the non-native conditions used. Thus, when *fruB* was overexpressed in either the triple mutant (TM; Fig 3C) or the WT genetic background (WT; Fig 3D), there was a significant increase in activity only when 2DG uptake was measured in the WT background. *fruB* overexpression did seem to stimulate 2DG uptake. Finally, when both *fruA* and *fruB* were simultaneously overexpressed using two different compatible plasmids, in either the TM or WT genetic background, the apparent stimulatory effects on Mtl, NAG, αMG and 2DG transport were similar to those observed when only *fruA* was overexpressed. Again, in this experiment, overexpression of *fruA*, *fruB*, or *fruA* and *fruB* did not cause apparent Tre and Gat transport to decrease.

To explain these results, we must consider that the overexpression of *fruA* and *fruB* caused by fructose induction was dissimilar from that obtained using the two compatible plasmids, pMAL and pZA31-PtetM2, which have different copy numbers. Thus, balanced and equimolar expression of *fruA* and *fruB* was achieved in the former case but not in the latter case. Additionally, both FruA and FruB interact with many proteins (see Introduction), which may influence the consequences of their high levels of expression. Nevertheless, the results confirm the suggestion that FruA is primarily responsible for the stimulation of Mtl, NAG, 2DG and αMG uptake, although FruB may play a lesser role.

## Operon induction properties using lacZ fusions

The three transport systems showing largest responses to fructose induction in wild type cells were the systems encoded by the mannitol (*mtl*), galactitol (*gat*) and mannose (*man*) operons (see Fig 2). Therefore, in order to determine if these effects reflected changes in operon transcription or EII activities, transcriptional *lacZ* fusions were constructed to operons encoding these three PTS systems (*mtlA-lacZ; gatY-lacZ* and *manXYZ-lacZ*). These were used in studies to determine the effects of fructose in the growth medium prior to transport rate determinations. The results are presented in Table 3. It can be seen that the presence of fructose in the growth medium, or the overexpression of specific *fru* operon genes, had no apparent effect on the induction of *mtl*, *gat*, and *man* operon expression. These results imply that the effects observed in Figs 2 and 3 reflect the enzyme II *activities* and not their *syntheses*. This conclusion is substantiated by the results obtained when the transphosphorylation reactions were studied (see Table 4 below).

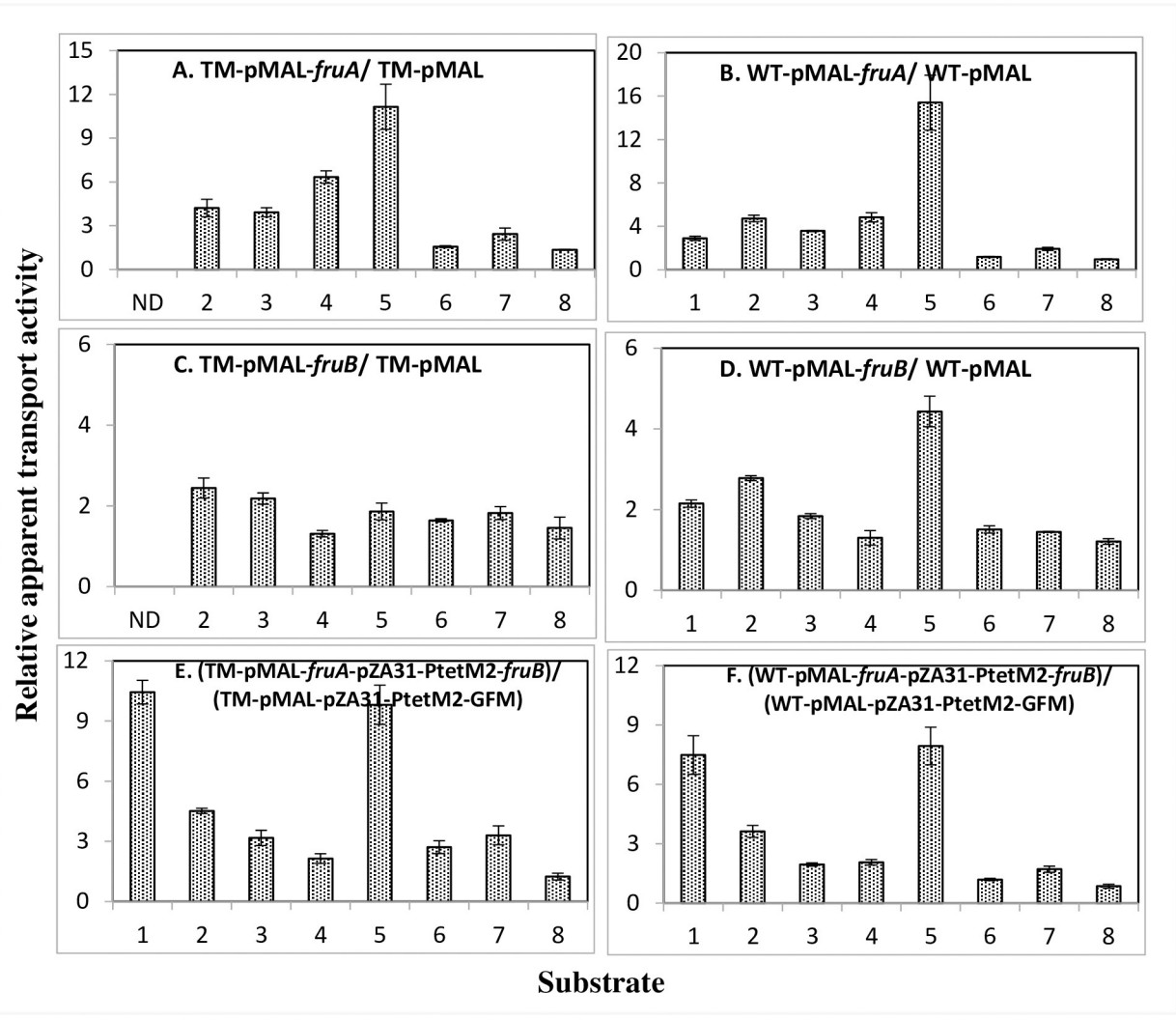

**Fig 3. Ratios of apparent transport activities for various sugars (PTS substrates and galactose) by wild type *E. coli* BW25113 (WT) and its** ***fruBKA*** **triple mutant (BW25113-*fruBKA:kn* (TM) over-expressing individual or combined *fruBKA* operon genes.** Ratios of apparent transport rates for [$^{14}$C]sugar uptake in wild type (WT) and *ΔfruBKA* mutant backgrounds over-expressing individual or combined *fruBKA* operon genes as presented at the tops of the figures: A. *fruA* in the TM strain. B. *fruA* in the WT parental strain. C. *fruB* in the TM strain. D. *fruB* in the WT parental strain. E. *fruA* and *fruB* in the TM strain. F. *fruA* and *fruB* in the WT parental strain. 1, Fructose; 2, Mannitol; 3, N-acetylglucosamine; 4, Methyl alpha glucoside; 5, 2-Deoxyglucose; 6, Trehalose; 7, Galactitol and 8, Galactose. The raw data for these plots are provided in supplementary materials (S6–S12 Tables). ND, not determined.

Summarizing, growth with or without fructose, or after overexpressing *fruA* and/or *fruB* had essentially no effect on the levels of the β-galactosidase activities of the *mtlA-*, *gatY-* and *manXYZ-lacZ* bearing strains. This clearly implied that the *mtl*, *gat* and *man* operons were not repressed or induced by the presence of fructose in a wild type or *ΔfruBKA* strain or by single or dual overexpression of *fruA* and *fruB* in the triple mutant, *ΔfruBKA*, genetic background.

## PEP-dependent phosphorylation of PTS sugars in vitro

Fig 4A presents the consequences of growth in the presence of fructose on the PEP-dependent phosphorylation of various PTS sugars. As expected, fructose phosphorylation activity increased dramatically, but mannitol and N-acetylglucosamine phosphorylation activities increased as well. Interestingly, galactitol phosphorylation decreased dramatically although

**Table 3. Ratios of the responses of *lacZ* fusion-bearing strains to various conditions and genetic backgrounds (S13–S15 Tables).** WT = wild type, BW25113; OE = overexpression; *ΔfruBKA* = deletion of the entire *fruBKA* operon, also called triple mutant, TM. All strains were grown in LB medium to which 0.2% fructose was added, only for the first two entries as indicated. The remaining entries reveal the consequences of the overexpression of specific *fru* genes or gene combinations on *lacZ*-fusion gene expression. In no case were the changes significant, suggesting that the presence of fructose in the growth medium or the overexpression (OE) of specific *fru* genes, or the entire *fru* operon, did not influence expression of the *mtl*, *gat*, or *man* operons.

| | *mtlA-lacZ* | *gatY-lacZ* | *manXYZ-lacZ* |
|---|---|---|---|
| WT, ± 0.2% fructose/WT (S13 Table) | 0.9 | 0.4 | 0.7 |
| *ΔfruBKA*, ± 0.2% fructose/*ΔfruBKA* (S13 Table) | 0.9 | 1.2 | 1.6 |
| WT *fruA* OE/WT (S14 Table) | 1.2 | 1.0 | 1.3 |
| WT *fruB* OE/WT (S14 Table) | 1.1 | 1.1 | 1.1 |
| *ΔfruBKA*, *fruA* OE/*ΔfruBKA* (S14 Table) | 1.1 | 1.1 | 1.0 |
| *ΔfruBKA*, *fruB* OE/*ΔfruBKA* (S14 Table) | 1.1 | 1.1 | 1.0 |
| *ΔfruBKA*, *fruA/fruB* OE/*ΔfruBKA* (S15 Table) | 1.0 | 0.9 | 1.1 |

phosphorylation of other PTS sugars were apparently not affected. Upon overexpression of the complete *fruBKA* operon (Fig 4B), or of the *fruA* and *fruB* genes co-expressed on compatible plasmids, similar results were obtained with increased phosphorylation activities of Fru > Mtl > NAG (Fig 3C and 3D). FruA or FruB overproduction alone had little or no effect, and the phosphorylation of other PTS sugars also showed little effect. Mannitol phosphorylation increased 4 to 6-fold, while phosphorylation of N-acetylglucosamine increased 2 to 3-fold, and phosphorylation of galactitol decreased to about one eighth. Inclusion of fructose, fructose 1-P, fructose 6-P or fructose 1,6-bisphosphate during the *in vitro* assay had essentially no effect on mannitol, N-acetylglucosamine or galactitol phosphorylation.

The *mtlA* gene was inactivated with a kanamycin resistance gene insertion or was deleted (*ΔmtlA*), and the crude extracts were compared with the wild type. Crude extracts were assayed for PEP-dependent [14C]mannitol phosphorylation after growth with or without fructose, and the activity was reduced to a few percent of the wild type level in the insertion or deletion mutant (S22 Table), showing that mannitol cannot be phosphorylated at an appreciable level by other PTS Enzyme II complexes under these conditions.

To further examine the effects of the FruA and FruB proteins on activation of other PTS Enzyme II complexes, recombinant FruB was purified to near homogeneity (S2 Fig), and it was added to crude extracts of the recombinant *ΔfruBKA* mutant ± overexpression of *fruA*. The results, presented in Table 5, show that purified FruB had a FruA-dependent activating

**Table 4. Effects of overexpressing *fruA* and varying amounts of purified FruB on the transphosphorylation activities of tested enzyme II complexes.** Results are expressed as the ratios of enzyme activities in the presence relative to the absence of purified FruB for the different preparations. The raw data are presented in S25 and S26 Tables. In all cases, neither FruA nor FruB influenced the transphosphorylation activities for the 5 sugar-specific Enzyme II complexes assayed.

| Radioactive sugar | Relative enzyme activity (OE FruA/TM) | Relative enzyme activity (EII plus purified FruB /EII alone) | | | | |
|---|---|---|---|---|---|---|
| | | TM-pMAL-*fruA* | | | | TM-pMAL |
| | | Purified FruB (µg) | | | | Purified FruB (µg) |
| | | 0.24 | 0.37 | 1.48 | 1.85 | 1.85 |
| | Value±SD | Value±SD | Value±SD | Value±SD | Value±SD | Value±SD |
| Mannitol | 1±0.14 | 1.01±0.1 | 0.98±0.11 | 0.87±0.05 | 0.8±0.08 | 1.07±0.13 |
| N-Acetylglucos-amine | 0.8±0.14 | 1.05±0.09 | 0.98±0.08 | 1.05±0.24 | 1.15±0.15 | 1.1±0.17 |
| Trehalose | 0.5±0.02 | 1.06±0.09 | 0.92±0.01 | 0.94±0.06 | 0.96±0.06 | 1.07±0.08 |
| Methyl alpha glucoside | 1.2±0.04 | 0.97±0.04 | 1.01±0.02 | 1.04±0.01 | 0.97±0.09 | 1.02±0.08 |
| 2-Deoxyglucose | 1.2±0.01 | 1.13±0.13 | 1.12±0.2 | 1±0.05 | 1.02±0.16 | 1.03±0.13 |

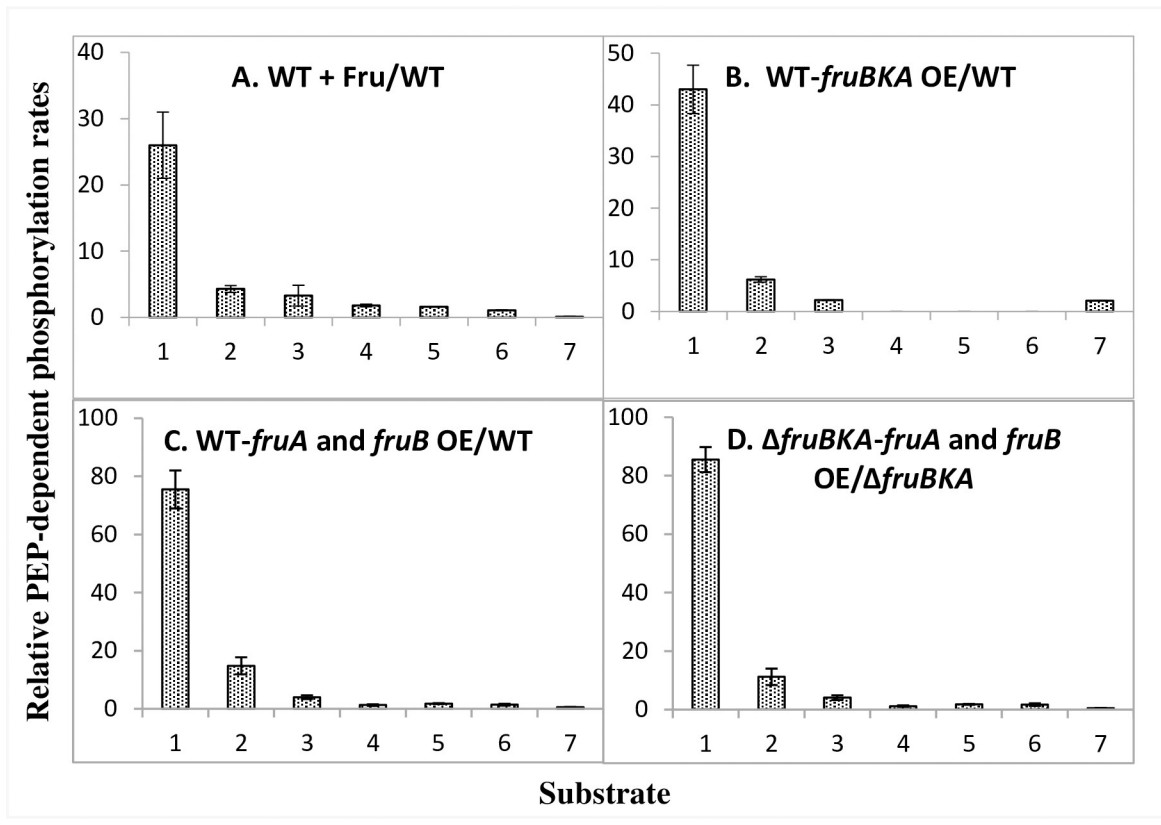

**Fig 4. PEP-dependent phosphorylation of various radioactive PTS sugars by crude extracts of wild type and Δ*fruBKA* strains of *E. coli* following either induction with fructose or overexpression of *fruBKA* or *fruA* and *fruB*.** Ratios of the PEP-dependent phosphorylation rates for various radioactive PTS sugars by crude extracts of wild type and Δ*fruBKA* strains of *E. coli*. A. wild type cells induced by growth in LB + fructose (0.2%) compared to LB grown cells. B. Effect of the overexpression of the entire *fruBKA* operon on *in vitro* PEP-dependent sugar phosphorylation rates when cells were grown in LB medium. C. The consequences of the simultaneous overexpression of *fruA* and *fruB* in the WT background. D. The same as C except that the Δ*fruBKA* strain was used. 1, Fructose; 2, Mannitol; 3, N-acetylglucosamine; 4, Methyl alpha glucoside; 5, 2-Deoxyglucose; 6, Trehalose and 7, Galactitol. The raw data for these plots are presented in S16–S21 Tables. Other conditions and combinations of gene overexpression did not result in appreciable changes in activities (see S16–S21 Tables).

effect on [$^{14}$C]mannitol phosphorylation *in vitro* when PEP was the phosphoryl donor, regardless of whether membrane pellets were used from the WT or Δ*fruBKA* strain, or whether a crude extract was used from the mutant lacking the *fru* operon. Purified HPr did not stimulate mannitol phosphorylation.

Moreover, while mannitol phosphorylation responded dramatically to the inclusion of FruB in the assay mixture, a lesser activation was observed when N-acetylglucosamine was the

**Table 5. Effect of purified FruB or HPr on the PEP-dependent phosphorylation of [$^{14}$C]mannitol using membrane pellets (MP) of *E. coli* strain BW25113-pMAL-*fruA* and BW25113-*fruBKA:kn*-pMAL-*fruA* or crude extracts (Cr.Ext.) of strain BW25113-*fruBKA:kn*-pMAL-*fruA* as compared to their corresponding control strains BW25113-pMAL and BW25113-*fruBKA:kn*-pMAL.**

| Purified protein added | Activity ratio (Enzyme II plus FruB or HPr /Enzyme II alone) | | | | | |
|---|---|---|---|---|---|---|
| | MP-WT-pMAL | MP-WT-pMAL-*fruA* | MP- Δ*fruBKA* -pMAL | MP- Δ*fruBKA* -pMAL-*fruA* | Cr.Ext-Δ*fruBKA* -pMAL | Cr.Ext-Δ*fruBKA* -pMAL-*fruA* |
| FruB (0.23 μM) | 1.5 | 8.6 | 1.2 | 8.1 | 1.2 | 7.5 |
| HPr (0.55 μM) | 2.0 | 2.3 | 1.7 | 2.3 | 1.5 | 1.5 |

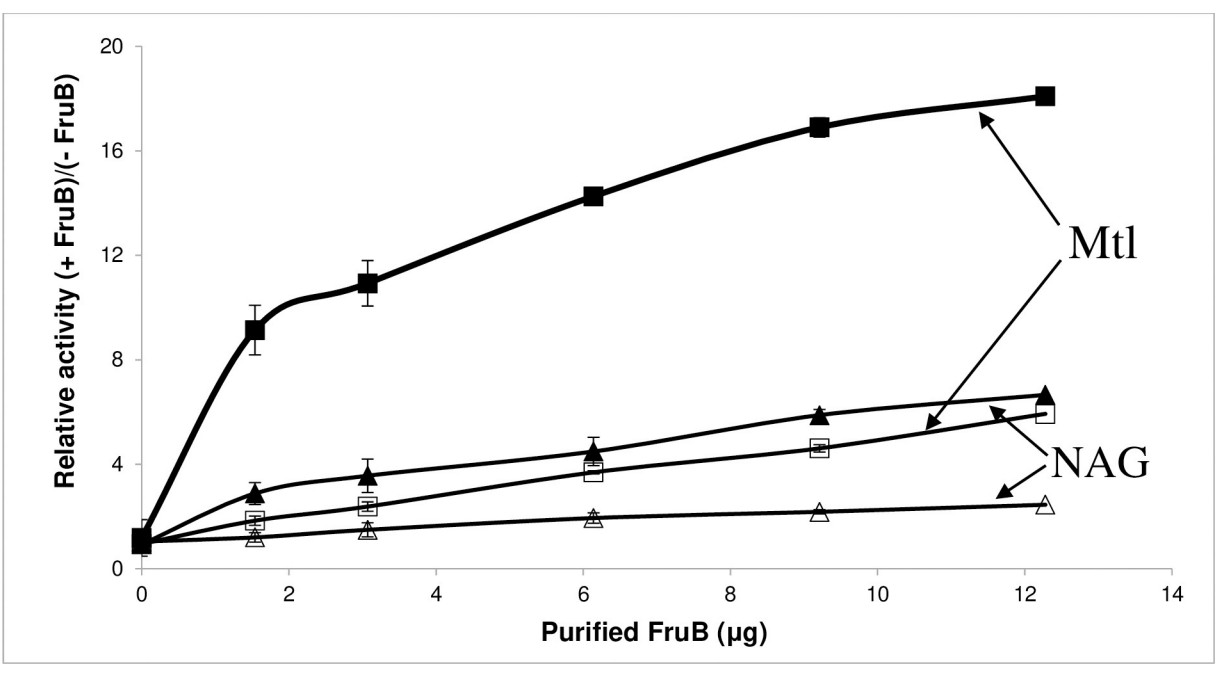

**Fig 5. Effect of purified FruB on PEP-dependent phosphorylation activities of crude extracts of the recombinant triple mutant *E. coli* strain BW25113-*fruBKA:kn*-pMAL-*fruA* (TM-pMAL-*fruA*, closed symbols) as compared to the BW25113-*fruBKA:kn*-pMAL (TM-pMAL, open symbols), assaying phosphorylation of [$^{14}$C]PTS sugars, mannitol (Mtl, squares) and N-acetylglucosamine (NAG, triangles).** The raw data are presented in S23 Table.

sugar substrate (Fig 5). Activation by FruB but not HPr showed that FruB activation is not due to the activity of the FPr domain of FruB but is dependent on the presence of FruA (Table 5).

The crude extract of *E. coli* strain BW25113-*mtlA:kn* or BW25113Δ*mtlA* grown in LB+0.2% fructose was prepared and used to demonstrate the absence of mannitol phosphorylation by FruA (S22 Table). The results eliminate the possibility that stimulation of mannitol phosphorylation was due to cross substrate specificity causing mannitol phosphorylation by EII$^{Fru}$.

### Effect of a high-speed supernatant (HSS), derived from *E. coli* cells with *fruBKA* overexpressed, on the *in vitro* PEP-dependent phosphorylation activities of membrane pellet preparations with overexpressed *fruA* or *galP*

In view of the *in vitro* phosphorylation results described above and presented in Table 5 and Fig 5, a high-speed supernatant [(HSS), supplying all of the soluble proteins of the PTS (Enzyme I, HPr and FruB)] from a strain overexpressing the *fruBKA* operon, was examined for the effects on phosphorylation activities of membrane pellet preparations overexpressing either *fruA* or *galP* (Fig 6 and S24 Table).

Increased expression of *fruB* in a high speed supernatant (HSS) from cells overexpressing *fruBKA*, gave a large increase (about 56-fold) of fructose phosphorylation as expected, but mannitol phosphorylation increased 9.1-fold, while N-acetylglucosamine phosphorylation increased 2.7-fold for the overexpressed *fruA* membrane pellets (Fig 6A). Overexpressed *galP* membrane pellets did not show activation (Fig 6B; S24 Table). Interestingly, inhibition of [$^{14}$C]galactitol phosphorylation observed in *fruA* overexpressed membrane pellets was similarly observed in *galP* overexpressed membrane pellets. These results suggest that while the activation of mannitol (and to a lesser extent, N-acetylglucosamine) phosphorylation *in vitro* is

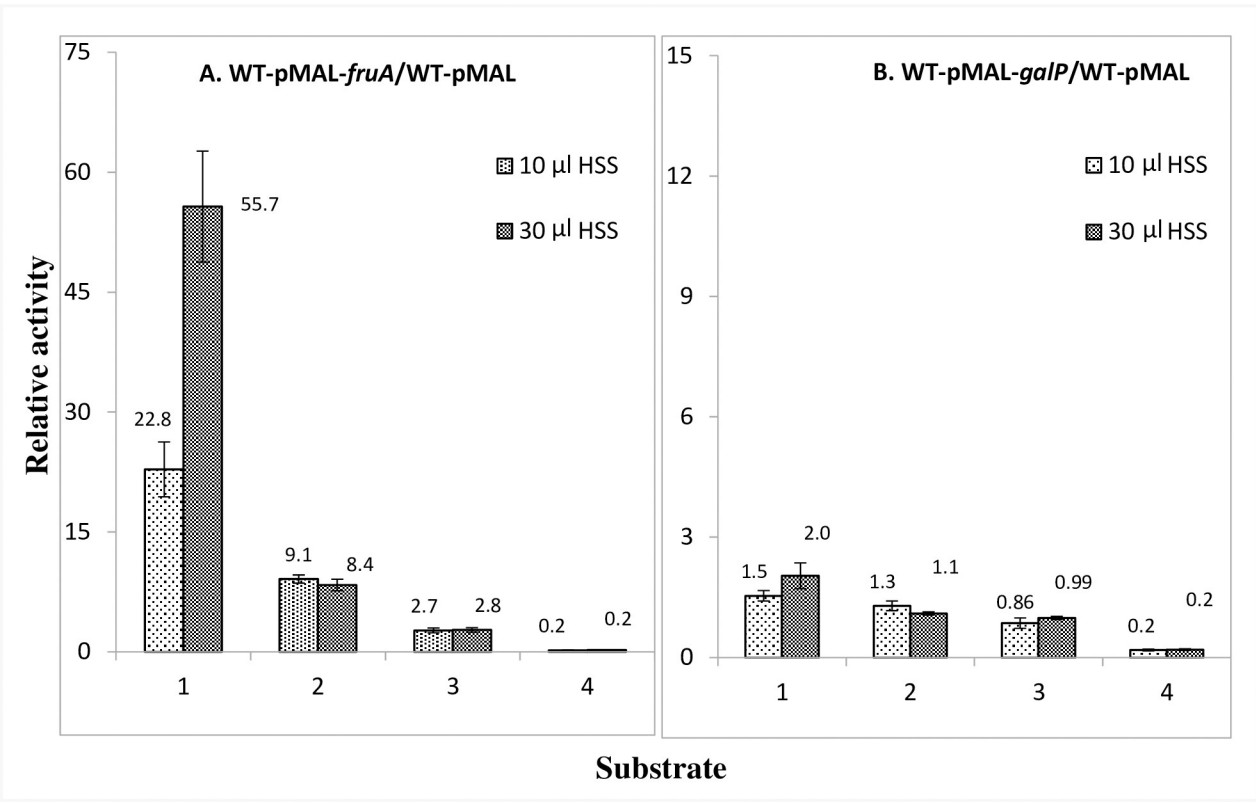

**Fig 6.** Effects of using a HSS from lysed *E. coli* BW25113 *chs kn:T:Ptet-fruBKA* cells, overexpressing the *fruBKA* operon, on the PEP-dependent phosphorylation activities of enzymes II in membrane pellets (MP) of *E. coli* BW25113 overexpressing *fruA* (A) or *galP* (B) for PTS sugars: 1, Fructose; 2, Mannitol; 3, N-acetylglucosamine and 4, Galactitol. An aliquot of an 8 h HSS from *E. coli* BW25113 *chs kn:T:Ptet-fruBKA* (WT OE *fruBKA* operon) was used as a source of soluble PTS enzymes for the measurement of PEP-dependent phosphorylation activities. The raw data are presented in S24 Table.

specific to the fructose PTS enzymes, FruA and FruB, the inhibition of galactitol phosphorylation is not as specific.

### Effects of *fruA* overexpression and purified FruB on sugar-P:sugar transphosphorylation reactions

In contrast to the PEP-dependent phosphorylation reactions catalyzed by the PTS, the sugar-phosphate:sugar transphosphorylation reactions depend only on the Enzyme IIB-IIC complex (as well as IID for the mannose system), not on Enzyme I, HPr or the Enzyme IIA proteins. Therefore, these reactions were studied as controls. The phosphorylation of several sugars (mannitol, N-acetylglucosamine, trehalose, methyl α-glucoside and 2-deoxyglucose) were assayed in the presence of either FruA (due to the overexpression of the *fruA* gene) alone or FruA and variable quantities of purified FruB (0.24, 0.37, 1.48 and 1.85 μg per reaction mixture). As shown in Table 4, in no case was there a significant effect (>50%) on the transphosphorylation reactions examined (see S25 & S26 Tables for the raw data).

These transphosphorylation reactions do not depend on protein:protein interactions, except for the mannose system, in which the IIAB$^{man}$ protein must interact with the IIC$^{man}$-IID$^{man}$ complex, and in this case, it appears that all constituents of the system are associated in the membrane. In all other cases examined, the IIB and IIC domains are present in a single polypeptide chain (see Table 1). Thus, it appears likely that the effects of varying the expression

level(s) of one or more PTS IIC/IIB component(s) on the *in vivo* transport and *in vitro* phosphorylation activities of other Enzyme II complexes of different specificities is due to facilitation or inhibition of protein-protein interactions within and between the Enzyme II complexes of the PTS. Thus, the results confirm the results obtained using the *lacZ* fusions and suggest that turnover of the EIIs does not influence the *in vivo* transport or *in vitro* PEP-dependent sugar phosphorylation results. FruA and FruB do not promote or inhibit synthesis of other PTS Enzymes. The effects appear to be on the transport and PEP-dependent phosphorylation *activities* of these enzyme transporters, not on their *syntheses* or sugar-P:sugar transphosphorylation activities.

## Discussion

As noted in the introductory section of this paper, the PTS, found only in prokaryotes, both bacteria and archaea, but not in eukaryotes so far examined, is complex, both in structure and function [2–11]. It consists of energy-coupling phosphoryl transfer proteins (Enzyme I, HPr, and the sugar-specific IIA and IIB proteins or protein domains) as well as the PTS sugar transporters, the IIC constituents of the systems that catalyze transfer of the phosphoryl moieties of the IIB~P proteins to the incoming sugars [3, 4, 6] (see Introduction and Table 1). However, it has also been shown that the PTS serves as a chemoreception system [38–41] and regulates carbon, nitrogen and energy metabolism [42–44]. Prior to the present study, the enzyme IIC proteins had not been shown to function as parts of regulatory networks involving direct protein-protein interactions within the membrane.

In a global interactome study involving both soluble and integral membrane proteins in *E. coli* K12 [1], we reported that many integral membrane proteins interact with each other, and particularly prominent, were those involving the fructose-specific Enzyme IIC proteins of the PTS (see Table 2). Because these PTS enzyme-transporters can be assayed *in vitro* as well as *in vivo*, we investigated the consequences of these interactions. Most striking was the observation that many of the PTS permeases interact with the fructose PTS (FruA/FruB), more than with any other PTS Enzyme II complex (Table 2). However, the biochemical and physiological significance of these interactions was not known. As a result, we carried out the molecular genetic, biochemical and physiological studies reported here, revealing that the activities of the Enzyme IICs of several different PTS sugar-specific transporters are enhanced by their interactions with FruA (the fructose Enzyme IIBC), but that others are inhibited by the corresponding interactions. These effects were large, as great as 20-fold, and far in excess of the standard errors of the measurements. The results also showed that regulation of the uptake of PTS sugars *in vivo* does not correlate *quantitatively* with regulation of sugar phosphorylation *in vitro*. This confirms conclusions of previous studies suggesting that although the two processes, transport and phosphorylation, are coupled under normal conditions *in vivo*, the rate limiting steps for phosphorylation *in vitro* are different from those for sugar uptake *in vivo* [45–47]. This can be explained, for example, if cell lysis partially disrupts the relevant protein-protein interactions.

The level of the fructose Enzyme II complex in the membrane proved to determine the extent of activation or inhibition of the target Enzyme II systems. Moreover, the magnitude and direction of regulation depended on the specific target PTS transport system under study. Thus, the glucose, mannose, mannitol and N-acetylglucosamine permeases exhibited enhanced *in vivo* sugar transport activities, and sometimes enhanced *in vitro* PEP-dependent sugar phosphorylation activities (but to differing extents), while the galactitol and trehalose systems showed inhibited activities (also to differing extents) when the fructose system was induced to a high level in wild type cells. Most of these effects were shown to occur either due to *fruBKA* operon induction by the presence of fructose during growth of the wild type cells used

for the assay, or to *fruA* and/or *fruB* gene overexpression in the absence of fructose in the growth medium. We found that the presence of fructose during assay *in vivo* or *in vitro* was unimportant for this regulation, except for induction of the regulating system as noted above. In fact, exhaustive studies failed to reveal a significant role for the presence of the sugar substrate or sugar-phosphate product of the activating or inhibiting Enzyme II complex on the extent of regulation, either during assay or during growth, except for induction of the regulating system.

As controls, first, the sugar-specific *transphosphorylation* activities of these PTS enzyme/transporters (dependent only on the presence of the IIBC domains, not on the presence of Enzyme I, HPr or the Enzymes IIA) [16–19, 48], and second, the rates of *synthesis* of the target PTS permeases, as measured using *lacZ* transcriptional fusions, were shown not to be affected. These observations clearly suggested that the activating or inhibiting effects observed were due to the consequences of direct protein-protein interactions within the membrane, only on the *activities*, not the syntheses, of the target enzymes II. The fact that the transphosphorylation reactions of these EIIs were not decreased by high level expression of II[Fru] additionally suggested that these EIIs did not undergo degradation or turnover.

It seemed reasonable, but was not proven, that regulation depended on interactions of the target PTS permeases with the energy coupling proteins of the PTS as well as the regulating Enzyme II. It should be noted, however, that the interactions might also influence the stabilities of the target Enzymes II, although this seems less likely due to the remarkable stability of the membrane-embedded EIIs as well as the uninfluenced transphosphorylation activities. We thus suggest that specific *protein-protein interactions* (PPIs) within the cytoplasmic membrane regulate the activities of PTS permeases in a physiologically meaningful way that may contribute to and enhance the importance of the hierarchy of preferred PTS sugars as previously demonstrated when studying catabolite repression [2, 49, 50]. Since these effects could be demonstrated in the absence of a functional fructose-transporting system, we suggest that it is the direct PPIs, and not the activities of the regulating proteins, that determine the regulatory effects. Thus, it is the *levels* of the regulating system that seem to determine the degrees of activation or inhibition of the target Enzyme II complexes.

While these conclusions appear valid for the *in vivo* transport results, the same seems to be true of the PEP-dependent sugar phosphorylation results, although this is less certain as the regulatory effects were substantially smaller. In this regard, both the lack of any regulatory effect on the IIBC-dependent *in vitro* transphosphorylation reactions and target PTS gene transcription, as well as the lesser regulatory effects *in vitro* compared to *in vivo*, strengthen the suggestion that complex formation involving PTS proteins plays an important role, and that cell disruption may have dissociated some of these interactions.

It is interesting to note that in a previous publication, we suggested that the fructose Enzyme IIABC complex was the primordial system, and that all other PTS Enzymes II arose later during evolution of the more complex PTS [20, 51]. The original evidence for this suggestion has been summarized in the Introduction. It is relevant to the present studies that only the fructose Enzyme II has a duplicated *IIB domain* (IIB') that functions in protein-protein interactions and not in phosphoryl transfer [21]. It is possible that this "extra" domain plays a role in the regulatory interactions documented in this report. Moreover, we have demonstrated the biochemical significance of the large number of protein-protein interactions for the fructose PTS, as compared with those for any other sugar-specific Enzyme II complex, again consistent with the suggestion that the fructose PTS was the primordial system.

The observations cited above, explaining the functions of the interactions of the fructose PTS with other Enzyme II complexes, appear to be applicable in principle to other Enzymes II as well as to other (non-PTS) types of transport systems. Thus, in preliminary studies, we found that several Enzymes II, in addition to the fructose Enzyme II, interact with other

Enzymes II (see Table 2), having activating or inhibiting effects that depend both on the level of the inhibiting Enzyme II complex and apparently, on the hierarchical position of the target systems under study. Moreover, we found that high level expression of genes encoding the Enzymes II of the PTS influence the activities of certain non-PTS integral membrane transporters in vivo, and vice versa. These observations lead to the suggestion that interactions of many integral membrane transport proteins (and possibly a variety of integral membrane enzymes) occur in the plane of the membrane, influencing their activities, possibly in a physiologically significant and relevant way. The experiments reported here therefore serve as guides to direct further experimentation aimed at defining the consequences of intra-membrane protein-protein interactions. Future studies will be required to determine the extent to which these interactions occur and prove to be physiologically important.

## Experimental procedures

### Constructions of bacterial two-hybrid plasmids

The Euromedex bacterial two-hybrid system (Euromedex, Cat # EUK001), consisting of two compatible plasmids, pUT18 and pKNT25, were designed to detect protein-protein interactions *in vivo*. pUT18 encodes Ap resistance and carries the ColE1 origin and the P*lac* promoter driving T18 domain of cAMP synthase of *Bordetella pertussis*, while pKNT25 encodes Kn resistance and carries the p15A origin and the P*lac* promoter driving T25 domain of cAMP synthase. To determine any possible interactions between two target proteins, one gene was fused to the N-terminus of T18 in pUT18 while the other was fused to the N-terminus of T25 of pKNT25. These two plasmids were co-transformed into an *E. coli* strain lacking its own cAMP synthase gene *cyaA*. In the case in which the two target proteins closely interacted, thereby bringing together the T18 and T25 domains, the *E. coli* cells synthesized cAMP, thereby exhibiting unique phenotypes that can be assessed.

*fruA*, *fruB*, *gatC*, *nagE*, *treB*, *mtlA*, *galP*, *lacY* and *pheP* were amplified by PCR from BW25113 genomic DNA using carefully designed oligonucleotides (S28 Table), digested with appropriate restriction enzymes, and then ligated into the same sites of pUT18 individually. In each of these resultant plasmids, the target structural gene, with no stop codon, is inserted immediately upstream of the N-terminus of T18 in pUT18, creating a single hybrid gene that encodes the target protein at the N-terminus and the T18 domain at the C-terminus. These plasmids are denoted pUT18-*fruA*, pUT18-*fruB*, pUT18-*gatC*, pUT18-*nagE*, pUT18-*treB*, pUT18-*mtlA*, pUT18-*galP*, pUT18-*lacY* and pUT18-*pheP*, respectively (S27 Table).

Similarly, *fruA*, *fruB*, *mtlA* and *nagE* were individually fused to the N-terminus of the T25 domain in pKNT25. Each of these resultant plasmids carries a single hybrid gene encoding the target protein at the N-terminus and the T25 domain at the C-terminus. These plasmids are denoted pKNT25-*fruA*, pKNT25-*fruB*, pKNT25-*mtlA* and pKNT25-*nagE*, respectively (S27 Table).

To test the possible interaction between two proteins, a recombinant pUT18 plasmid and a recombinant pKENT25 plasmid were co-transformed by electroporation into the *E. coli* BTH101 strain deleted for *cyaA*. The transformants were inoculated onto MacConkey agar plates supplemented with maltose, kanamycin (Kn), ampicillin (Ap) and IPTG. The plates were incubated at 30°C for up to 48 h. The appearance of red colonies indicated positive interactions between the target proteins. The intensity of red color was proportionally correlated to the level of protein-protein interactions.

### Constructions of fruA, fruB, fruBKA and mtlA deletion mutants

Strains (and plasmids) and DNA oligonucleotides used in this study are described in Supplementary S27 and S28 Tables, respectively. The deletion mutants of *fruA*, *fruB*, *fruBKA* and

*mtlA* were generated from the parental strain (*E. coli* K-12 strain BW25113) using a standard method as described in [52]. Briefly, to construct each mutant, a kanamycin resistance gene (*kn*), flanked by the FLP recognition site (FRT), amplified from the template plasmid pKD4 using a pair of specific mutation oligos (S28 Table), was first substituted for the target gene or operon. Where indicated, the *kn* gene was subsequently eliminated (leaving an 85-bp FRT sequence) using plasmid pCP20 that bears the FLP recombinase. Replacements of the target genes/operons by the FRT-flanking *kn* gene and subsequent removal of the *kn* gene were confirmed by colony PCR and DNA sequencing, yielding deletion mutants Δ*fruA*, Δ*fruB*, Δ*fruBKA* and Δ*mtlA*, respectively (S27 Table).

## Construction of the Ptet driven fruBKA strain

Using plasmid pKDT:P*tet* [53] as template, the DNA region containing the *kn* gene and the P*tet* promoter was PCR amplified using a specific primer pair, P*tetfruB*-P1/P*tetfruB*-P2 (S28 Table). The PCR products were integrated into the BW25113 chromosome to replace the *fruBKA* promoter (P$_{fruBKA}$ between the 123$^{th}$ nucleotide and the 1$^{st}$ nucleotide relative to the translational start point of *fruB*). This chromosomal integration led to P*tet*-driven *fruBKA* expression (strain BW_P*tet*-*fruBKA*) (S27 Table).

## Construction of overexpression plasmids

pMAL-p2X [54] was used to overexpress various cytoplasmic and transport proteins in *E. coli*. This plasmid carries a ColE1 origin, *lacI*q and a strong IPTG inducible promoter P*tac*, driving *malE* gene expression (useful in making N-terminal fusions for protein purification purposes). To make a control plasmid (carrying P*tac* but not *malE*), pMAL-p2X was digested with *Bgl*II (+435 to +440 relative to the start site of *malE*) and *Sal*I (located downstream of *malE* in the multiple cloning site region) and then re-ligated, yielding pMAL-empty which is the same as pMAL-p2X except that most of *malE* has been removed.

The structural regions of *fruA* (encoding the fructose uptake transporter), *fruB* (encoding the fructose-specific PTS multiphosphoryl transfer protein) and *galP* (encoding the galactose: H$^+$ symporter), were PCR amplified from BW25113 genomic DNA (using specific pairs of primers as indicated in S28 Table), digested with *Nde*I and *Bam*HI, and then ligated into the same sites of pMAL-p2X individually. In each resultant recombinant plasmid, the target structural gene (no promoter and no 5' UTR) is substituted for *malE* in pMAL-p2X, and its expression is exclusively under the control of the IPTG inducible promoter P*tac*. These plasmids were referred to as pMAL-*fruA*, pMAL-*fruB* and pMAL-*galP*, respectively (S27 Table).

## Chromosomal PmtlA-lacZ, Pman-lacZ and PgatY-lacZ Transcriptional Fusions

The *mtlA* promoter region (-386 bp to +57 bp relative to the *mtlA* translational start point), the *man* promoter region (-234 bp to +54 bp relative to the *manX* translational start point) and the *gatY* promoter region (-204 bp to +42 bp relative to the *gatY* translational start point), each plus a stop codon at the 3' end, were amplified from BW25113 genomic DNA. These DNA fragments, referred to as promoter regions (each containing a promoter region, the first 14 to 19 codons plus a stop codon at the end) were then inserted between the *Xho*I and *Bam*HI sites of the plasmid pKDT [53], yielding the plasmids pKDT_P*mtlA*, pKDT_P*man* and pKDT_P*gatY*, respectively. In each of these newly made plasmids, an *rrnB* terminator (*rrnB*T) is present between the *kn* gene and the downstream cloned promoter. The DNA fragments containing the *kn* gene, *rrnB*T and those of the promoter regions (plus the first 14 to 19 codons followed by a stop codon) were PCR amplified from the above plasmids and individually

integrated into the chromosome of MG1655 carrying the seamless *lacY* deletion [53] to replace *lacI* and P*lacZ* but not the 5' UTR of *lacZ*. All of these chromosomal integrations were confirmed by colony PCR and subsequent DNA sequencing analyses. These promoter-*lacZ* fusions were individually transferred to BW25113 by P1 transduction, yielding strains BW_P*mtlA-lacZ*, BW_P*man-lacZ* and BW_P*gatY-lacZ*, respectively. Similarly, these reporters were transferred to Δ*fruBKA*, yielding strains Δ*fruBKA*_P*mtlA-lacZ*, Δ*fruBKA*_P*man-lacZ* and Δ*fruBKA*_P*gatY-lacZ*, respectively (S27 Table). In each of these reporter strains, the promoter of interest drives the first 14 to 19 codons of the target gene followed by the *lacZ*' 5' UTR and the structural gene.

### β-Galactosidase assays

The *E. coli* reporter strains were grown in 5 ml of media contained in 18 mm diameter glass test tubes under the same conditions as for the uptake experiments. During incubation, samples were removed for measurements of $OD_{600nm}$ and β-galactosidase activities after being appropriately diluted.

To measure β-galactosidase activities, 0.8 ml of Z-buffer containing β-mercaptoethanol (2.7 μl/ml) and SDS (0.005%) were mixed with 0.2 ml of sample and 25 μl of $CHCl_3$ in test tubes [55]. The tubes were vortexed twice (each time for 10 seconds) at a constant speed and incubated in a 37˚C water bath until equilibration. A 0.2 ml aliquot of ONPG substrate (4 mg/ml) was then added to each test tube. When yellow color developed, the reaction was stopped by adding 0.5 ml 1 M $Na_2CO_3$ followed by vortexing. After that, the reaction mixtures were centrifuged, and the absorbance values of the supernatants were measured at 420 nm and 550 nm. A control tube was run in parallel using diluted or undiluted LB broth instead of the test sample. β-galactosidase activity was expressed in Miller units = [(OD420-1.75XOD550)/[(sample volume in ml) X time in min X $OD_{600nm}$] X 1000 X dilution factor [56].

### Culture conditions for uptake of radioactive substrates

A fresh culture (100 μl of the test strain) was used to inoculate 5 ml of LB (in the case of wild type or mutant strains) or LB plus either 100 μg/ml ampicillin for pMAL plasmid harboring strains, or 100 μg/ml ampicillin plus 25 μg/ml chloramphenicol for strains harboring the pMAL/pZA31$P_{tet}$M2-GFM plasmids. The 18 mm diameter tubes were incubated in a shaking water bath at 250 rpm and 37˚C for about 6 h. Aliquots of 100 μl of the cultures were used to inoculate 50 ml of LB/5 mM $MgSO_4$ with or without 0.2% fructose (in the cases of wild type or mutant strains, and strains containing the $P_{tet}$ promoter in their chromosomes), or 50 ml of LB plus either 100 μg/ml ampicillin or 100 μg/ml ampicillin plus 25 μg/ml chloramphenicol (in the cases of the dual plasmid-harboring strains mentioned above) contained in 250 ml conical flasks, which were incubated in a shaking water bath at 250 rpm and 37˚C for 8 h (in the cases of wild type and mutant strains and strains containing the $P_{tet}$ promoter in their chromosomes) or 6 h, followed by a 2 h induction period with IPTG at a final concentration of 0.2 mM (recombinant plasmid harboring strains). During the IPTG induction period, fructose (0.2%) was included in certain experiments, and the shaking rate and the incubation temperature were lowered to 200 rpm and 30–32˚C, respectively. The cells were harvested by centrifugation in a Sorvall centrifuge at 4˚C and 10,000 rpm for 20 min. The cell pellets were washed 3x (in the absence of added sugar in the growth medium) or 4x (when a sugar was present in the growth medium). Each wash was with 35 ml of 50 mM Trizma-maleate buffer, pH 7, containing 5 mM $MgCl_2$. The pellets were resuspended in the same buffer to $OD_{600nm}$ values of 0.5, 0.25 or 0.125 for radioactive substrate uptake experiments.

## Radioactive substrate uptake

For the uptake assays, the stock solution of each radioactive substrate was used at 5 μCi/μmole for $^{14}$C while radioactive tritium was used at 30 μCi/μmole, and in all cases, the substrate concentration of the stock solution was 1 mM. Whenever stated, uptake assays were performed in the presence and absence of certain added non-radioactive sugars. For all uptake assays, the prepared bacterial suspensions of the test strains were used within a time period not exceeding 12 h. The assay mixtures contained 900 μl of a bacterial cell suspension of specified $OD_{600nm}$, 50 μl of 1 M arginine (pH 7), and 20 μl of the 1 mM stock radioactive substrate. The volume was brought to 1 ml with a cold sugar solution and/or 50 mM Trizma-maleate buffer, pH 7, containing 5 mM $MgCl_2$.

The final concentration of the radioactive sugar substrate in the assay mixture was 20 μM for uptake assays and 10 μM for phosphorylation assays unless otherwise stated. The uptake assays were carried out in a shaking water bath at 37˚C for 5–10 min followed by immediate filtration of withdrawn samples (100–250 μl) through 0.45 micron membrane filters under vacuum. The filters containing cells were washed 3x with cold 50 mM Trizma-maleate buffer, pH 7, containing 5 mM $MgCl_2$ before being dried under infra-red lamps. Each dried filter was mixed with 10 ml Biosafe NA solution in scintillation vials, and the radioactivity, expressed as counts per min (CPM), was measured in a Beckman scintillation counter. For normalization of values among samples for different test strains, the radioactivity was expressed as CPM/0.1 $OD_{600nm}$/0.1 ml/min.

## Culture conditions for phosphorylation assays

A fresh culture of the test strain was used as an inoculum which was prepared by inoculating 5 ml contained in an 18 mm diameter test tube or 50 ml contained in 250 ml conical flasks of LB medium without or with the appropriate sugar (e.g., fructose at 0.2%) and antibiotic(s) (100 μg/ml ampicillin (Ap) for pMAL plasmid harboring strains, or 100 μg/ml ampicillin plus 25 μg/ml chloramphenicol (Cm) for strains harboring the pMAL/pZA31P$_{tet}$M2-GFM plasmids). The flasks were incubated at 37˚C, either in a gyratory shaking water bath at 250 RPM for the 250 ml flasks, or in a shaking incubator at 275 RPM for the 2 l flasks for 8 h (wild type, mutant strains and strains containing the chromosomal P$_{tet}$ promoter), or for 6 h plus a 2 h induction period (recombinant strains). Induction was carried out by adding IPTG to a final concentration of 0.2 mM at 32˚C and 200 RPM. The cells were harvested by centrifugation at 4˚C, washed 3x with cold modified M63, and then re-suspended in about 7 ml (pellets from 50 ml cultures) or 30 ml (pellets from 1 l cultures) of modified M63 containing 5 mM DTT. The prepared cell suspensions were disintegrated by three passages through a French press at 12,000 PSI. The resultant cell lysates were centrifuged at 10,000 RPM for 10 min at 4˚C in a SORVALL centrifuge, and the supernatants produced, termed crude extracts (CE), were either used directly for PEP-dependent phosphorylation assays of PTS enzymes II or subjected to ultracentrifugation in a Beckman centrifuge, Ti-70 titanium rotor, for 2 h at 40,000 RPM. The pellets produced were re-suspended in an appropriate volume of modified M63/5 mM DTT to give the membrane pellets (MP) that were used for PEP-dependent and transphosphorylation assays of PTS enzymes. The 2 h high speed supernatants (HSS) produced were kept overnight on ice at 4˚C and then subjected to one or more 2 h centrifugation cycles under the previously mentioned conditions. This was occurred once, twice or thrice to get 4, 6 or 8 h HSSs, which were used as sources of soluble PTS enzymes for PEP-dependent phosphorylation assays. The repeated centrifugations were necessary to remove the last traces of the membranous Enzymes II [16, 54].

## PTS phosphorylation assays

These assays were performed as previously described by Aboulwafa and Saier, 2002 [57]. For the PEP-dependent reactions, Enzyme II preparations were either membrane pellets (MP), or crude extracts (CE), and the following assay mixtures were used: 50 mM potassium phosphate buffer (pH 7.4), 10 μM [$^{14}$C]sugar, or [$^{3}$H]sugar (in the case of galactitol), 5 mM phosphoenolpyruvate (PEP), 12.5 mM MgCl$_2$, 25 mM KF and 2.5 mM dithiothreitol (DTT). For the sugar-P-dependent reactions (transphosphorylation), the same was applied except that membrane pellets only were used as the enzyme II source, PEP was replaced by a sugar-P (usually at a 1000-fold higher concentration than that of the radioactive sugar unless otherwise stated), and the pH of the potassium phosphate buffer was 6.0. The sugar phosphates used were: glucose-6-phosphate with [$^{14}$C]methyl α-glucoside or [$^{14}$C]2-deoxyglucose; fructose-1-phosphate with [$^{14}$C]fructose, N-acetylglucosamine-6-phosphate with [$^{14}$C]N-acetylglucosamine, trehalose-6-phosphate with [$^{14}$C]trehalose, and mannitol-1-phosphate with [$^{14}$C]mannitol. The radioactive sugar/sugar phosphate concentrations were 10 μM/10 mM except that they were 50 μM/5 mM and 25 μM/5 mM for the [$^{14}$C]methyl α-glucoside and [$^{14}$C]2-deoxyglucose phosphorylation assays, respectively. The resin used to separate [$^{14}$C]sugar from [$^{14}$C]sugar-phosphate was Dowex® 1X8, chloride form, 50–100 mesh (Sigma-Aldrich). After 4x washing of columns containing resin with deionized water, the 1 M lithium chloride eluate (9 ml), containing the radioactive sugar-P, was mixed with 10 ml of Biosafe II solution for radioactivity measurements in a Beckman scintillation counter. Cold sugar, sugar phosphate, the soluble purified FruB enzyme or the purified HPr protein was incorporated in the reaction mixtures at the specified concentrations.

## Protein purification

The His-tagged FruB from an *E. coli* strain BW25113-*fruBKA*:*kn*-pMAL-*fruB* culture, overexpressing the *fruB* gene or the equivalent strain overexpressing the *ptsH* (HPr) gene, was purified by nickel affinity chromatography. The cells were grown in LB and induced for 2 h with 0.2 mM IPTG as stated before for the preparation of cells for the transport assays. The cells were harvested by centrifugation, washed 3x with modified M63 and subjected to 3 cycles of freezing at -20˚C and thawing before being lysed with B-PER Complete Bacterial Protein Extraction Reagent (from Thermo Scientific, Cat # 89821) following the manufacturer's instructions. His-FruB or His-HPr in the cell extract was purified using HisPur™-Ni-NTA Superflow Agarose (from Thermo Scientific, Cat # 25214) following the manufacturer's instructions. The eluates, produced from the agarose columns, containing purified His-FruB or His-HPr, were desalted by passage through pre-equilibrated (with modified M63/5 mM DTT) Zeba™ Spin Desalting columns (Thermo Scientific, Cat # 89894). Each Zeba column contained 10 ml of size exclusion resin MWCO 7000 Da. The desalted preparations were concentrated using Amicon ultra centrifugal filter devices, MWCO 5000 Da, Cat # UFC 900502. The purity of the resultant preparation was checked by SDS-PAGE as described by Aboulwafa and Saier, 2011 [54]. Phenylmethylsulfonyl fluoride (PMSF) was incorporated into the purified preparation at 1 mM before being aliquoted and stored at -80˚C.

## Determination of protein concentrations

The protein concentrations were determined using the Biorad colorimetric protein assay (Cat. #500–0006) with bovine serum albumin as the standard protein.

## Materials

All radioactive sugars were purchased from New England Nuclear (NEN) Corp. or American Radiolabeled Chemicals (ARC). Nonradioactive compounds were from commercial sources,

usually from the Sigma Chem. Corp. unless otherwise noted, and were of the highest purity available commercially.

## Supporting information

**S1 Table. Effect of induction with fructose on the uptake of radioactive compounds as indicated below by the wild type *E. coli* strain BW25113 (WT).** *E. coli* WT was grown in LB or LB plus 0.2% fructose with 5 mM $MgSO_4$ in both media. All [$^{14}$C]substrates were used at 20 μM, final concentration, each of 5 μCi/μmole except for [$^3$H]galactitol which was used at 30 μCi/μmole (see Experimental Procedures). Many values reported here and in subsequent tables have been rounded off.
(DOCX)

**S2 Table. Effect of growth with fructose on the uptake of radioactive substrates as indicated below by the triple mutant *E. coli* strain BW25113-*fruBKA:kn* (TM).** The *E. coli* TM was grown in LB or LB plus 0.2% fructose with 5 mM $MgSO_4$ in both media. All radioactive substrates were used at 20 μM, each containing 5 μCi/μmole $^{14}$C except for [$^3$H]galactitol which was used at 30 μCi/μmole. These concentrations and specific activities were used throughout these studies for transport assays (see Experimental Procedures).
(DOCX)

**S3 Table. A.** Effect of growth with fructose on the uptake of [$^{14}$C]substrates as indicated below except for [$^3$H]galactitol by the triple mutant *E. coli* strain BW25113-*fruBKA:kn* (TM) as compared to the wild type strain BW25113 (WT). *E. coli* strains were grown in LB plus 0.2% fructose and 5 mM $MgSO_4$. **B.** Effect of mutations in the *fruBKA* operon on the uptake of [$^{14}$C]compounds by the triple mutant *E. coli* strain BW25113-*fruBKA:kn* (TM) as compared to the wild type strain BW25113 (WT), both grown in LB medium.
(DOCX)

**S4 Table. A.** Effect of deletion of *fruA* on the uptake of [$^{14}$C]compounds by the *E. coli* strain BW25113Δ*fruA* (WTΔ*fruA*) as compared to the wild type strain BW25113 (WT). *E. coli* strains were grown in LB/0.2% fructose/5 mM $MgSO_4$. **B.** Effect of deletion of *fruA* on the uptake of [$^{14}$C]compounds by the *E. coli* strain BW25113Δ*fruA* (WTΔ*fruA*) as compared to the wild type strain BW25113 (WT), both grown in LB medium.
(DOCX)

**S5 Table. A.** Effect of deletion of *fruB* on the uptake of [$^{14}$C]compounds by the *E. coli* strain BW25113Δ*fruB* (WTΔ*fruB*) as compared to the wild type strain BW25113 (WT), both grown in LB medium plus 0.2% fructose/5 mM $MgSO_4$. **B.** Effect of deletion of *fruB* on the uptake of [$^{14}$C]compounds by the *E. coli* strain BW25113Δ*fruB* (WTΔ*fruB*) as compared to the wild type strain BW25113 (WT), both grown in LB medium.
(DOCX)

**S6 Table. Effect of overexpression of *fruA* on the uptake of [$^{14}$C]compounds by the recombinant triple mutant *E. coli* strain BW25113-*fruBKA:kn*-pMAL-FruA (TM-pMAL-*fruA*) as compared to the BW25113-*fruBKA:kn*-pML (TM-pMAL) strain.**
(DOCX)

**S7 Table. Effect of overexpression of *fruA* on the uptake of [$^{14}$C]compounds by the recombinant *E. coli* strain BW25113-pMAL-*fruA* (WT-pMAL-*fruA*), as compared to the BW25113-pMAL (WT-pMAL) strain, both grown in LB medium.**
(DOCX)

**S8 Table. Effect of overexpression of *fruB* on the uptake of [$^{14}$C]compounds by the recombinant triple mutant *E. coli* strain BW25113-*fruBKA:kn*-pMAL-*fruB* (TM-pMAL-*fruA*) as compared to the BW25113-*fruBKA:kn*-pML (TM-pMAL) strain, both grown in LB medium.**
(DOCX)

**S9 Table. Effect of overexpression of *fruB* on the uptake of [$^{14}$C]compounds by the recombinant *E. coli* strain BW25113-pMAL-*fruB* (WT-pMAL-*fruB*), as compared to the BW25113-pMAL (WT-pMAL) strain, both grown in LB medium.**
(DOCX)

**S10 Table. Effect of co-overexpression of *fruA* and *fruB* carried on two separate compatible plasmids on the uptake of [$^{14}$C]compounds by the recombinant triple mutant *E. coli* strain BW25113-*fruBKA:kn*-pMAL-*fruA*-pZA31-*PtetM2*-*fruB* (TM-pMAL-*fruA*-pZA31-*PtetM2*--*fruB*) as compared to the BW25113-*fruBKA:kn*-pMAL-pZA31-*PtetM2*-*GFM* (TM-pMAL-pZA31-*PtetM2*-*GFM*) strain, both grown in LB medium.**
(DOCX)

**S11 Table. Effect of co-overexpression of *fruA* and *fruB* carried on two separate compatible plasmids on the uptake of [$^{14}$C]compounds by the recombinant *E. coli* strain BW25113-pMAL-*fruA*-pZA31-*PtetM2*-*fruB* (WT-pMAL-*fruA*-pZA31-*PtetM2*-*fruB*) as compared to the BW25113-pMAL-pZA31-*PtetM2*-*GFM* (WT-pMAL-pZA31-*PtetM2*-*GFM*) strain, both grown in LB medium.**
(DOCX)

**S12 Table. Effect of overexpression of the *fruBKA* operon under *Ptet* promoter control on the uptake of [$^{14}$C]compounds by the recombinant *E. coli* strain BW25113-*Chs kn:T:Ptet-fruBKA* (WT-*Ptet-fruBKA*) as compared to the BW25113 (WT) strain, both grown in LB medium.**
(DOCX)

**S13 Table. Effect of *fruBKA* operon induction with fructose on the expression of certain PTS transporters in *E. coli* using *lacZ* transcriptional fusions.** Values in the last column were calculated relative to the control without induction. A negative sign indicates a decrease in apparent expression level. All values are within experiment error.
(DOCX)

**S14 Table. Effect of overexpression of the fructose transporter (FruA) gene (*fruA)* and the gene of its soluble partner FruB (*fruB)* on the expression of certain PTS transporters in *E. coli* using *lacZ* transcriptional fusions.** Values in the last column were calculated relative to the control without induction. A negative sign indicates a decrease in expression level.
(DOCX)

**S15 Table. Effect of overexpression of some membrane transporter genes on the expression of other PTS transporters in *E. coli* using *lacZ* transcriptional fusions. Values in the last column were calculated relative to the control without induction. A negative sign indicates a decrease in expression level.**
(DOCX)

**S16 Table. Effect of induction by fructose on PEP-dependent phosphorylation of some PTS sugars by the crude extract preparations of the wild type *E. coli* BW25113 (WT) and its triple mutant BW25113-*fruBKA:kn* (TM).** Induction was conducted with 0.2% fructose in LB medium; ND, not determined.
(DOCX)

**S17 Table. Effect of overexpression of *fruA* on phosphorylation of fructose, mannitol, N-acetylglucosamine and galactitol by the membranous fractions of the wild type *E. coli* BW25113 strain (WT). A 30 μl aliquot of an 8 h HSS of the WT strain was used as a source of the soluble PTS enzymes.**
(DOCX)

**S18 Table. Effect of overexpression of *fruA* or *fruB* on PEP-dependent phosphorylation of PTS sugars by crude extracts of the recombinant wild type *E. coli* strains BW25113-pMAL-*fruA* and BW25113-pMAL-*fruB* as compared to the control strain BW25113-pMAL.**
(DOCX)

**S19 Table. Effect of separate overexpression of *fruA* or *fruB* on the PEP-dependent phosphorylation of PTS sugars by the crude extracts of recombinant triple mutant *E. coli* strains BW25113-*fruBKA*:*kn*-pMAL-*fruA* and BW25113-*fruBKA*:*kn*-pMAL-*fruB* as compared to the control strain BW25113-*fruBKA*:*kn*-pMAL.**
(DOCX)

**S20 Table. Effect of simultaneous overexpression of *fruA* and *fruB* (carried on two separate compatible plasmids in the wild type *E. coli* BW25113 strain or its triple mutant BW25113-*fruBKA*:*kn*) on the PEP-dependent phosphorylation of some PTS sugars by crude extracts of strains BW25113-pMAL-pZA31-*PtetM2-GFM* (WT control), BW25113-pMAL-*fruA*-pZA31-*PtetM2-fruB* (WT O.E. *fruA-fruB*), BW25113-*fruBKA*:*kn*-pMAL-pZA31-*PtetM2-GFM* (TM-control) and BW25113-*fruBKA*:*kn*-pMAL-*fruA*-pZA31-*PtetM2-fruB* (TM-O.E *fruA.fruB*).**
(DOCX)

**S21 Table. Effect of overexpression of the *fruBKA* operon by changing its native promoter to *Ptet* on the PEP-dependent phosphorylation of mannitol, N-acetylglucosamine and galactitol by the membranous fraction of the wild type *E. coli* BW25113.**
(DOCX)

**S22 Table. Testing PEP-dependent phosphorylation of mannitol by a crude extract of an *E. coli* strain with an inactivated (BW25113-*mtlA*:*kn* strain) or deleted (BW25113Δ*mtlA* strain) *mtlA* gene.** A crude extract of *E. coli* strain BW25113-*mtlA*:*kn* or BW25113Δ*mtlA* grown in LB + 0.2% fructose was prepared and used for testing the mannitol phosphorylation by FruA.
(DOCX)

**S23 Table. Effect of purified FruB on the PEP-dependent phosphorylation of PTS sugars by the crude extracts of recombinant triple mutant *E. coli* strain BW25113-*fruBKA*:*kn*-pMAL-*fruA* (Triple Mutant; TM-pMAL-*fruA*) and BW25113-*fruBKA*:*kn*-pMAL (TM-pMAL) strain.**
(DOCX)

**S24 Table. Effect of using a HSS of lysed *E. coli* BW25113 Chs *kn*:T:Ptet-*fruBKA* cells overexpressing the *fruBKA* operon on the PEP-dependent phosphorylation of fructose, mannitol, N-acetylglucosamine and galactitol by the wild type *E. coli* BW25113 overexpressing *fruA* or *galP*.**
(DOCX)

**S25 Table. Effect of overexpression of *fruA* on transphosphorylation of PTS sugars by the membranous fraction of recombinant triple mutant *E. coli* strain BW25113-*fruBKA*:*kn*-**

pMAL-*fruA* as compared to control strain BW25113-*fruBKA:kn*.
(DOCX)

**S26 Table. Effect of purified FruB on transphosphorylation of PTS sugars by the recombinant triple mutant *E. coli* strain BW25113-*fruBKA*:kn-pMAL (TM-pMAL) and its *fruA* overexpressing strain BW25113-*fruBKA*:kn-pMAL-fruA (TM-pMAL-*fruA*).**
(DOCX)

**S27 Table. Strains and plasmids used in this study.**
(DOCX)

**S28 Table. Oligonucleotides used in this study.**
(DOCX)

**S1 Fig. Results from the use of a bacterial two hybrid system for testing interactions of FruA and FruB with other membrane proteins.** pKNT25-*fruA* (expression in low copy plasmid pKNT25) against pUT18-*gatC*, pUT18-*nagE*, pUT18-*treB*, pUT18-*mtlA*, and pUT18-*fruB* (expression in high copy plasmid pUT18); and pUT18-*fruB* against pKNT25-*mtlA* and pKNT25-*nagE* are presented. As reported by Babu et al., 2018, FruA interacts with other membrane proteins. In this study, the interactions of FruA could be confirmed for GatC, NagE, TreB and MtlA, and the interactions of FruB could be detected and reproduced for MtlA and NagE. The interactions of FruB, the soluble partner, with FruA can be considered as a positive control. Note that the interactions of FruB with MtlA and NagE appear to be substantially weaker than those with FruA.
(DOCX)

**S2 Fig. SDS PAGE of the purified FruB preparation from the recombinant *E. coli* BW25113-*fruBKA*:kn-pMAL-*fruB* strain.** M, molecular weight markers; Lanes 1 and 2, two purified FruB preparations.
(DOCX)

## Author Contributions

**Conceptualization:** Mohammad Aboulwafa, Milton H. Saier, Jr.

**Data curation:** Mohammad Aboulwafa.

**Formal analysis:** Mohammad Aboulwafa, Milton H. Saier, Jr.

**Funding acquisition:** Milton H. Saier, Jr.

**Investigation:** Mohammad Aboulwafa, Zhongge Zhang.

**Methodology:** Mohammad Aboulwafa, Zhongge Zhang.

**Project administration:** Milton H. Saier, Jr.

**Resources:** Milton H. Saier, Jr.

**Validation:** Mohammad Aboulwafa.

**Writing – original draft:** Mohammad Aboulwafa, Milton H. Saier, Jr.

**Writing – review & editing:** Mohammad Aboulwafa, Milton H. Saier, Jr.

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
