## [Decision Letter · Decision Letter 0]

10 Sep 2019

PONE-D-19-16979

Protein:Protein Interactions in the Cytoplasmic Membrane Influencing Sugar Transport and Phosphorylation Activities of the E. coli Phosphotransferase System.

PLOS ONE

Dear Prof. Saier Jr.,

Thank you for submitting your manuscript to PLOS ONE. We invite you to submit a revised version of the manuscript that addresses the points raised during the review process. You may confirm that the comments of the panel of reviewers are very complete, positive and constructive. In general,  their opinion, and mine, is that the manuscript is technically complete, well written, and that the topic is highly relevant. In any case, one reviewer suggested major revision, while the two remaining suggested a minor revision, but I agree that most of their major concerns can be addressed without further experiments. However, there is at least one experiment that should be performed before publication. The steady state levels of key proteins of your study should measured in crude extracts or membrane pellets, as suggested by the reviewers, to fully support important conclusions of your manuscript.

We would appreciate receiving your revised manuscript by Oct 25 2019 11:59PM. To enhance the reproducibility of your results, we recommend that if applicable you deposit your laboratory protocols in protocols.io, where a protocol can be assigned its own identifier (DOI) such that it can be cited independently in the future. For instructions see: http://journals.plos.org/plosone/s/submission-guidelines#loc-laboratory-protocols

We look forward to receiving your revised manuscript.

Kind regards,

Hernâni Gerós, PhD

Academic Editor

PLOS ONE

Journal Requirements:

Additional Editor Comments (if provided):

Reviewers' comments:

Reviewer's Responses to Questions

**Comments to the Author**

1. Is the manuscript technically sound, and do the data support the conclusions?

Reviewer #1: Partly

Reviewer #2: Yes

Reviewer #3: Yes

2. Has the statistical analysis been performed appropriately and rigorously? 

Reviewer #1: N/A

Reviewer #2: Yes

Reviewer #3: Yes

3. Have the authors made all data underlying the findings in their manuscript fully available?

Reviewer #1: Yes

Reviewer #2: No

Reviewer #3: Yes

4. Is the manuscript presented in an intelligible fashion and written in standard English?

Reviewer #1: Yes

Reviewer #2: Yes

Reviewer #3: Yes

5. Review Comments to the Author

Reviewer #1: The present manuscript presents experiments that show that modification of protein expression of the fructose-specific phosphotransferase transport (PTS) system has a positive or negative effect on the apparent activity of other sugar-specific PTS systems in Escherichia coli. Based on previously published global interactome data, which show that soluble and integral membrane elements of multicomponent PTS systems interact with each other, with the the fructose PTS FruA and FruB components forming the most extensive PTS interactome network, they further analyze the basis of this phenomenon.

They use different physiological conditions related to induction of the fructose specific PTS and different strains of E. coli, genetically deleted for FruA or/and FurB, as well as, plasmid-driven overexpression of the Fur components, to investigate how other PTS systems are affected. The authors use basically two functional methods to draw their conclusions; in vivo measurements of radiolabeled sugar accumulation in intact cells, and in vitro enzymatic assays measuring sugar-specific transphosphorylation apparent activities. They additionally use β-galactosidase transcriptional fusions to test whether the de novo synthesis of the affected PTS components is altered in a FurA/B PTS-dependent manner. Based on these assays they conclude that the observed positive or negative effect of FurA/B PTS expression on the activity of other sugar-specific PTS systems take places via protein-protein interaction within the PM, and go on speculating that this is a physiologically meaningful cellular mechanism by which prokaryotes finely regulate PTS-dependent sugar uptake in complex and constantly changing environments.

Main conclusion

This is a very interesting scientific story, which happens to be familiar to me, as in my laboratory we have also come across phenomena of dominant negative or positive effects related to apparent transport activities, caused by the overexpression or absence of a specific transporter. Our observations concern fungal transporters, which makes the whole story broader and probably concern the fine and dynamic balancing of PM transporter composition in all kinds of cells. So, the findings described worth publication in PlosOne, but only after some important issues are clarified, conclusion partly modified and some extra experiments are made. In conclusion the text/figures should be partly re-written, conclusions modified, and a western blot analysis performed. Please see details below.

Points to addressed

1. The tile of the manuscript is not justified by the results presented. (Protein:Protein Interactions in the Cytoplasmic Membrane Influencing Sugar Transport and Phosphorylation Activities of the E. coli Phosphotransferase System). Protein-protein interactions are apparently the reason of the dominant negative or positive effects observed but this is not formally shown.

2. Introduce a cartoon showing the components of different PTS systems in the introduction. The existing figure is not friendly to the reader. Please use full names when abbreviations appear for first time in the text. What is PEP for example?

3. The first two paragraphs of Results introduction recapitulate published results from the same group. It can be significantly reduced with reference in Table 2 or entirely deleted.

4. Confirmation of the interactome results using a bacterial two hybrid system: I cannot see any figure related to this, except S2. It is not clear whether the 2-hybrid system is partly performed here or partly previous publication? Why S2 appears before S1?

5. Having the figure legends interrupting the text in annoying. Either you need both figure/figure legends within the text, or both at the end.

6. In Table S1 explain better what WT and WT-Fr means. I guess the latter means induction by fructose while the former stand for non0induced conditions. One also guess that each experiment is carried out twice, as for each substrate two values are given in all cases, no?

Please, explain abbreviations in all figure legends.

7. Uptakes shown describe ‘apparent’ transporter activity or better ‘apparent accumulation of radiolabelled substrate’, not transport activities per se. Moreover, they are carried out for 5-10 min which definitively describes steady state accumulation and not rate of uptake, despite the results given per min. To my knowledge uptake rates are liner of much shorter times than 5-10 min.,

8. The accumulation of some sugars (e.g. Trehalose and Galactinol) increases significantly (up to 9-fold) in the TM compared to WT upon fructose addition (Table S3). However, the addition of fructose makes a 2-fold difference (Table 2). Please explain this point better in the text.

9. There are some significant deviations in different experiments (S1-S3). Please comment on those.

10. Why S Tables are labelled S4, S5 and S4’, S5’, and not simply with consequent numbers?

11. Most of the data appear in 26 supplementary tables! This makes the manuscript difficult to read and little elegant. I suggest that more clear and composite main figures (with several panels) should highlight the most important results. These should appear in the main text. I am in general against very long supplementary material. The main article should be read without the need for supplements, which are only there to reinforce main findings.

12. A main experiment missing is a western for measuring steady state levels of all components that are affected. Modification of o expression in specific PTS could have an effect on the translocation of PTS in the PM via the translocon complex or in the stability of PTS due to displacement in non-physiological PM microdomains. These possibilities should be discussed.

13. The in vitro PEP-dependent phosphorylation assays are not convincing as they do not lead to a general conclusion. Moreover, I cannot see how crude extracts of membrane proteins can have an effect. My doubts are further enhanced by the fact sugar-P:sugar transphosphorylation assays had no effect by fruA overexpression or purified fruB.

14. “The effects appear to be on the transport and PEP-dependent phosphorylation

activities of these enzyme transporters, not on their syntheses or sugar-P:sugar

transphosphorylation activities” No.

The effect is on apparent activities which is most easily explained by imbalances in protein-protein interactions, including modification in translocation in the PM, stability and/or localization in specific PM microdomains.

15. The first part of the discussion should be deleted, is a repetition of the introduction and results. Also lines 478-507 should be deleted, as they are a repetition of information given ealrler.

Reviewer #2: The manuscript of Aboulwafa et al. addresses the biochemical and physiological significance of interactions of the fructose PTS transporter, consisting of FruA and FruB, with other PTS transporters, including glucose, mannitol, N-acetylglucosamine, galactitol and trehalose transporters. The authors show that: 1) the previous interactome results of FruA and FruB (Nat Biotechnol. 2018; 36:103-112) can be reproduced using a bacterial two-hybrid system, 2) through FruA and FruB, fructose stimulates transport of glucose, mannitol, and N-acetylglucosamine but inhibits transport of galactitol and trehalose, 3) overproduction of FruA and/or FruB (in the absence of fructose) also enhances transport activities of glucose, mannitol, and N-acetylglucosamine (but does not affect significantly those of galactitol and treharose), 4) either induction with fructose or overexpression of FruAB increases the PEP-dependent phosphorylation activities of mannitol and N-acetylglucosamine but decreases that activity of galactiol, and 5) such conditions do not affect transcription of the mannitol, galactitol, and mannose operons. On the basis of the present and previous results, the authors conclude that the fructose PTS transporter regulates the activities of several different PTS transporters through protein-protein interactions. Although mechanistic aspects remain for the future, the findings presented here provide a new perspective on the regulation of sugar transport in bacteria. Overall, this manuscript is clearly written and presents good experimental data to support the major claims of this study. However, I make some minor comments and questions that the authors will address the following issues, and suggest additional experiments, which will hopefully improve the manuscript.

1. Although the authors often describe about mannose transport activity (Lines 33, 251, and 280), the data is not presented anywhere.

2. Add sugar substrate(s) of each transport system in Table 1. This would be helpful for readers.

3. Why does overproduction of FruA and/or FruB not decrease treharose and galactitol transport activities? This is incompatible with the result obtained from mutational analysis of fruA and/or fruB (Fig. 1).

4. Lines 350-351, “FruB activation is not due to the activity of the FPr domain of FruB”. I do not agree with this claim because the authors do not analyze effect of this domain on sugar phosphorylations.

5. Line 372, galP. Add a brief description of galP.

6. Lines 41-42, “the rates of synthesis and protein levels in the membrane of the target PTS permeases were not altered”, Line 288, “the enzyme II activities and not their synthesis”, or Line 475, “only on the activities, not the synthesis or turnover”. To rule out the possibility that the fructose system affects synthesis or turnover of the target PTS proteins, the authors should check expression levels of one or more of these proteins in crude extracts and membrane pellets used for in vitro experiments as well as bacterial culture used for in vivo experiments. Only testing transcriptional activities (Table 3) is insufficient to support the authors’ claim.

Reviewer #3: In their manuscript “protein:protein interactions in the cytoplasmatic membrane influencing sugar transport and phosphorylation activities of the E. coli phosphotransferase system” the authors carefully investigate new roles of FruA and FruB, which are components of the PTS for fructose uptake and phosphorylation. In a previous study among many other results potential interactions of FruA and FruB with further EII-proteins for uptake of other sugars were observed. In this manuscript the authors now follow up and carefully investigate by various approaches the role of FruA and FruB for the control of PTS mediated transport and phosphorylation of sugars via the previously identified potentially interacting EII proteins. The authors here demonstrate that the EII permeases for glucose, mannose, mannitol, and N-acetylglucosamine show a FruA and FruB dependent enhancement of transport activity, while for trehalose and galacitol EII-protein FruA and FruB mediated transport inhibition is observed. The here observed novel effects on the PTS mediated uptake of sugars are shown to exclusively depended on the presence of FruA and FruB and do not depend on fructose transport itself. The authors demonstrate that the positive effects by FruA and FruB on mannitol, galacitol, and mannose uptake are neither based on increased of transcription of the operons encoding the EII components for the uptake of these sugars nor altered transphosphorylation activities, but a stimulation of PEP-dependent phosphorylating activity mediated via protein:protein interactions. Taken together, the authors in this article provide novel insights into a new mode for the control of sugar uptake via the PTS in E. coli and thereby provide further good indications, that this control occurs via protein:protein interactions. The authors finally discuss based on these additional role of FruA and FruB the evolution of PTS systems in bacteria.

The article is very interesting and well written, the experiments fully support the conclusions by the authors. Just some minor concerns might be addressed before publication.

Line 41: change to fructose

Line 58: Maybe instead of putting this scheme in the text an additional graph would be usefull, as some legend might be helpful. Thereby also the situation and names of genes/proteins for the uptake of the other PTS substrates investigated in this manuscript e.g. trehalose can be introduced, facilitationg the understanding of names e.g. used in table 2.

Line 189/Table 2: why not include the results from the bacterial 2 hybrid assay now presented in the supplementary data file also in table 2?

Table S1 and S2: For some sugars 2 and for galactose 3 series of measurements are provided. For the latter the uptake rates between the first two and the third series a highly different – maybe some explanation would be helpful.

Table S1 – S5: Addition of Fructose is abbreviated by writing e.g. “WT-Fru”, I suggest to replace this by “LB + Fru” (and accordingly for the control by “LB”).

Fig S2 (page 31 of supplementary data file) change “mtlA” to “MtlA” and “nagE” to “NagE”

6. PLOS authors have the option to publish the peer review history of their article (what does this mean?). If published, this will include your full peer review and any attached files.

Reviewer #1: No

Reviewer #2: No

Reviewer #3: No

---

## [Author Response · Author response to Decision Letter 0]

4 Oct 2019

Reviewer #1 (R1)

 We were pleased to read that R1 has observed similar phenomena with fungal transport activities, particularly because there is no PTS in fungi. We have since come to the same conclusion with respect to secondary carriers in E. coli (see Discussion section). As noted by R1, “Our observations concern fungal transporters, which makes the whole story broader and probably concern the fine and dynamic balancing of PM transporter composition in all kinds of cells. So the findings described [are] worth publication in PlosOne…”

Points to address:

1. We have inserted the word “Apparently” into the title as suggested. Similar changes have been made throughout the manuscript.

2. We have read through the entire manuscript to ensure that all abbreviations are defined when first used. The scheme shown on pg. 1 of the Introduction has been clarified by additional explanations in the text, both in the Abstract and the Introduction.

3. Table 2 and the first two paragraphs of the Results section summarize results published by Babu et al., 2018 (Ref. 1), as noted by R1, but the relevant observations were not presented concisely or explained in that paper as we have done here for the first time. Therefore, we feel this material should not be omitted from this manuscript. It would be difficult for the reader to find this information in Ref. 1. We also checked for redundancies with the Introduction and feel that most of the information presented is relevant and important; it reinforces the suggestions stated in the Introduction. However, the long section on PTS evolution in the Discussion section that overlapped with the Introduction has been deleted. Thus, although we have eliminated unnecessary details, as well as this major section, we feel that some of this information should be retained.

4. We have included the bacterial 2-hybrid-documented interaction results for FruA and FruB only in Figure S2 because this is only confirmatory of the results already published in Ref. 1 (although using a different method). It is therefore of secondary importance and does not represent novel information; hence we prefer to keep this information in the supplementary materials section.

5. We agree with R1; interrupting text paragraphs is not optimal. In our original submission, we tried to put figures and tables between paragraphs, and also tried to keep the tables or figures together with their legends. We have made sure that this was done in the revised manuscript.

6. Yes, R1 is correct: WT-Fru was meant to mean WT grown with Fructose. This has been changed to WT+Fru as suggested also by another reviewer. This has also been more clearly explained in the figure legends and text.

7. Uptake or transport activities have been changed to “apparent uptake or transport activities” in relevant places in the manuscript and Tables. Please refer to Aboulwafa et al., 2004 (Arch Microbiol (2004) 181 : 26-34, DOI 10.1007/s00203-003-0623-7) and Aboulwafa and Saier, 2002 (Research in Microbiology 153 (2002) 667-677) to see that apparent transport rate can often be determined within the time range 5-10 min for several of the tested substrates. Please note: “accumulation” is applicable for non-metabolizable compounds only, while rate of uptake plus metabolism is appropriate for other sugars.

8. The high ratios of accumulation of Tre and Gat by TM/WT is due to the lower activities of the WT after growth in the presence of fructose, not to a change in uptake by the TM. The results are fully consistent with our interpretation of the data.

R1 notes an apparent discrepancy between Tables S3 and 2. I assume R1 means Table S2 instead of Table 2? Table 2 only presents relevant interaction scores from Ref. 1.

In Table S2, only the TM strain is examined, and with this strain, there is no fructose PTS; hence, fructose has no effect on uptake. However, in Table S3, WT uptake activity is low, compared to the TM because of the inhibition of Gat and Tre uptakes by IIFru, and hence the ratio of TM+Fru (no effect)/WT+Fru (strong inhibition) is large. This is because only the fructose-induced WT shows inhibition by high levels of FruA on Gat and Tre uptakes. These experiments also establish that FruAB is responsible for the activation/inhibition of the other PTS transporters.

9. The reviewer is correct; there are significant deviations between experiments compared to the standard deviations for any one experiment conducted with the same cells, but we still attribute these to experimental error. This has been noted in the text.

10. As noted above, the S’ tables are controls for the corresponding S tables.

11. The supplementary tables provide the raw data upon which the figures and tables in the main manuscript are based. The relevant S tables for each text table are given in the text table legends. To include S tables in the text would introduce substantial redundancy and could confuse the reader as this would introduce a different style of presentation.

12. These possibilities are now discussed in the Discussion section as suggested.

13. We believe that the PEP-dependent in vitro phosphorylation results are significant. They confirm previous studies showing that cell disruption seems to (at least partially) uncouple sugar transport from phosphorylation. Values reported are much greater than the standard error of the measurements, and they show that in some cases, the in vitro effects observed are in the same direction as the in vivo effects although smaller in magnitude, possibly because cell disruption also disrupts protein-protein interactions. This has also been considered in the Discussion section of the revised manuscript.

14. We have qualified this statement (which we still believe to be essentially true) by mentioning the possibility of an effect on stability of the target transporters, although as noted above, and now stated in the revised manuscript, this seems less likely. Thus, the transphosphorylation reactions were not affected, suggesting that the target EIIs are stable. Moreover, it is known that EIIs are very stable in cell membranes (for days at RT and up to ~80oC for short time periods). This argument and these facts have now been presented in the revised text.

15. It is true that the first two paragraphs in the Discussion section in part paraphrase parts of the Introduction, but we believe this helps the reader to view the results in a fully comprehensive picture. We therefore prefer to retain this information. However, if acceptance for publication depends on its removal, we will certainly do so. Also, we have substantially shortened the Discussion section concerning PTS evolution because of the overlap with the Introduction.

Reviewer #2 (R2)

1. When we discuss mannose transport, we are talking about the results obtained with the non-metabolizable mannose analog, 2DG, which is a good substrate only of the mannose system. This point is now clarified in the revised manuscript (in lines 251 and 280 of the original manuscript) by noting that 2DG accumulation reflects the activity of the mannose PTS. This has also been clarified by putting the primary substrates into Table 1, as suggested.

2. Yes, we agree, and this has been done.

3. This is an interesting point. We believe that while induction by including fructose in the growth medium led to a full complement of regulatory effects, artificial overproduction of proteins from the gene(s) encoded on a synthetic plasmid did not. It caused the activating effects but did not reproduce the inhibitory effects on the Gat and Tre systems. We assume that this latter observation was due to the abnormal condition resulting from the presence of an overexpression plasmid, from which transcription/translation must occur. Possibly, these conditions (partially) prevent formation of the necessary association between FruA/FruB and the inhibited systems. This is mere speculation, but we have mentioned it in the Results section. This suggestion is also supported by studies on adenylate cyclase activation by the PTS which is easily demonstrable in whole cells and even toluenized cells, but never in a fully in vitro assay system.

4. Lines 350-351: We believe this statement is justified because the homologous HPr and FPr catalyze the same reaction, using the same mechanism, with the same specificity. If FPr were responsible for the activation, an excess of HPr should have had the same activating effect, but it did not. It had little or none; this is now more clearly pointed out.

5. Line 372, galP: a brief description of GalP has been added (as suggested) to the figure 2 legend.

6. Lines 41-42: R2 is correct, that we have only shown that synthesis (gene expression) is not affected, but turnover was not tested rigorously. This has been noted in the text as mentioned above and the statement in the abstract has also been accordingly modified. However, please note that the transphosphorylation results and other evidence now cited in the manuscript argue against the possibility that turnover plays a role.

Reviewer #3 (R3)

We thank R3 for the positive comments, particularly: “The authors of this article provide novel insights into a new mode of control” and “The experiments fully support the conclusions…”

Minor concerns:

Line 41: Thank you for catching this Typo! It has been corrected.

Line 58: As noted above, we have provided substantially more description in the text regarding this scheme and feel that these additions render it much more easily understandable. Similar schemes have been published in numerous publications concerning the PTS in the past.

Line 189/Table 2: This was addressed above. Since these results merely confirm the previously published results (Babu et al., 2017; Ref. 1) and do not represent an important conclusion with respect to the substance of this paper, we believe this figure optimally belongs in supplementary materials. The results now appear in S2 Figure.

Tables S1 and S2: This has also been addressed above. Please note that different strains and conditions were used for these and Tables S3 etc., accounting for the differences, excluding experimental error.

Tables S1-S5: We agree and have changed WT-Fru or WT-Fr to “WT + Fru”. The same has been done for other tables.

Figure S2 (pg. 31): The evidence for several relevant interactions is presented.

---

## [Decision Letter · Decision Letter 1]

14 Oct 2019

Protein:Protein Interactions in the Cytoplasmic Membrane Apparently Influencing Sugar Transport and Phosphorylation Activities of the E. coli Phosphotransferase System.

PONE-D-19-16979R1

Dear Dr. Saier Jr.,

We are pleased to inform you that your manuscript has been judged scientifically suitable for publication and will be formally accepted for publication once it complies with all outstanding technical requirements.

With kind regards,

Hernâni Gerós, PhD

Academic Editor

PLOS ONE

Additional Editor Comments (optional):

Reviewers' comments:

Reviewer's Responses to Questions

**Comments to the Author**

1. If the authors have adequately addressed your comments raised in a previous round of review and you feel that this manuscript is now acceptable for publication, you may indicate that here to bypass the “Comments to the Author” section, enter your conflict of interest statement in the “Confidential to Editor” section, and submit your "Accept" recommendation.

Reviewer #1: All comments have been addressed

Reviewer #3: All comments have been addressed

2. Is the manuscript technically sound, and do the data support the conclusions?

Reviewer #1: Yes

Reviewer #3: Yes

3. Has the statistical analysis been performed appropriately and rigorously? 

Reviewer #1: N/A

Reviewer #3: Yes

4. Have the authors made all data underlying the findings in their manuscript fully available?

Reviewer #1: Yes

Reviewer #3: Yes

5. Is the manuscript presented in an intelligible fashion and written in standard English?

Reviewer #1: Yes

Reviewer #3: Yes

6. Review Comments to the Author

Reviewer #1: All my points were satisfactorily addressed and the revised manuscript can be accepted fro publication

Reviewer #3: (No Response)

7. PLOS authors have the option to publish the peer review history of their article (what does this mean?). If published, this will include your full peer review and any attached files.

Reviewer #1: Yes: George Diallinas

Reviewer #3: No

---

## [Editor Report · Acceptance letter]

5 Nov 2019

PONE-D-19-16979R1 

Protein:Protein Interactions in the Cytoplasmic Membrane Apparently Influencing Sugar Transport and Phosphorylation Activities of the *E. coli* Phosphotransferase System. 

Dear Dr. Saier Jr.:

I am pleased to inform you that your manuscript has been deemed suitable for publication in PLOS ONE. Congratulations! Your manuscript is now with our production department. 

With kind regards,

on behalf of

Dr. Hernâni Gerós 

Academic Editor

PLOS ONE